# Noninvasive and reliable automated detection of spreading depolarization in severe traumatic brain injury using scalp EEG

Alireza Chamanzar [1,2✉], Jonathan Elmer[3], Lori Shutter [4], Jed Hartings[5] & Pulkit Grover [1,2✉]

## Abstract

**Background** Spreading depolarizations (SDs) are a biomarker and a potentially treatable mechanism of worsening brain injury after traumatic brain injury (TBI). Noninvasive detection of SDs could transform critical care for brain injury patients but has remained elusive. Current methods to detect SDs are based on invasive intracranial recordings with limited spatial coverage. In this study, we establish the feasibility of automated SD detection through noninvasive scalp electroencephalography (EEG) for patients with severe TBI.

**Methods** Building on our recent WAVEFRONT algorithm, we designed an automated SD detection method. This algorithm, with learnable parameters and improved velocity estimation, extracts and tracks propagating power depressions using low-density EEG. The dataset for testing our algorithm contains 700 total SDs in 12 severe TBI patients who underwent decompressive hemicraniectomy (DHC), labeled using ground-truth intracranial EEG recordings. We utilize simultaneously recorded, continuous, low-density (19 electrodes) scalp EEG signals, to quantify the detection accuracy of WAVEFRONT in terms of true positive rate (TPR), false positive rate (FPR), as well as the accuracy of estimating SD frequency.

**Results** WAVEFRONT achieves the best average validation accuracy using Delta band EEG: 74% TPR with less than 1.5% FPR. Further, preliminary evidence suggests WAVEFRONT can estimate how frequently SDs may occur.

**Conclusions** We establish the feasibility, and quantify the performance, of noninvasive SD detection after severe TBI using an automated algorithm. The algorithm, WAVEFRONT, can also potentially be used for diagnosis, monitoring, and tailoring treatments for worsening brain injury. Extension of these results to patients with intact skulls requires further study.

## Plain language summary

Physical injury to the brain, for example due to head trauma, may worsen over time, resulting in long-term disability or death. A spreading depolarization is a slowly spreading wave in the brain, which, if detected, can be used to predict worsening brain injuries. Current methods to detect spreading depolarizations require surgeries, which are risky and unlikely to be recommended to patients with mild brain injuries. In this work, we develop an automated monitoring technique for non-surgical, non-invasive detection of spreading depolarizations, called WAVEFRONT. We validated the performance of WAVEFRONT in 12 patients with severe brain injury. Our results demonstrate the feasibility of non-invasive detection of spreading depolarizations. Our approach can potentially help clinicians predict outcomes of brain injury patients, and tailor treatments accordingly.

[1] Electrical and Computer Engineering Department, Carnegie Mellon University, Pittsburgh, PA, USA. [2] Neuroscience Institute, Carnegie Mellon University, Pittsburgh, PA, USA. [3] Departments of Emergency Medicine, Critical Care Medicine and Neurology, University of Pittsburgh School of Medicine, Pittsburgh, PA, USA. [4] Department of Critical Care Medicine, Neurology and Neurosurgery, University of Pittsburgh School of Medicine, Pittsburgh, PA, USA. [5] Department of Neurosurgery, University of Cincinnati, Cincinnati, OH, USA. ✉email: achamanz@andrew.cmu.edu; pgrover@andrew.cmu.edu

This paper aims to address the question of whether spreading depolarization (SD) waves in the brain can be detected noninvasively. This is a long-standing question in the field of neurocritical care, with limited and contrasting reported results regarding the feasibility of SD detection using scalp electroencephalography (EEG). SDs are waves of neurochemical changes in the brain, which propagate slowly (1–8 mm/min) across the cortical surface and suppress neural activity[1–3]. SDs are caused by a breakdown in the ionic homeostasis across neuronal membranes[4]. Increasing evidence shows that SDs are associated with poor clinical outcomes in traumatic brain injuries (TBIs), strokes, and hemorrhages[4–9] and that this association is causal, such that the neurophysiological sequelae of SDs directly worsen secondary brain injury[9–15]. Recent studies have explored the potential of SD as a therapeutic target in subarachnoid hemorrhage (SAH)[9,16–18] and TBI[19,20]. Each year, around 69 million patients worldwide suffer from TBI[21] with 2.5 million patients in the United States[22]. Recent data on five-year outcomes for patients with moderate and severe TBI in the United States indicate that more than half of these patients experienced neurological worsening or death[23]. Therefore, detection of SD as a reliable biomarker and potential therapeutic target may help improve clinical outcomes.

Previous attempts to detect SDs using EEG have had limited success. In[24], Hartings et al. reported amplitude depressions associated with 81% of the intracranially detected SDs through visual inspection of scalp EEG recordings in severe TBI patients. In an earlier work[25], Drenckhahn et al. reported similar positive results for malignant stroke and SAH patients. In another recent work, some SD-type patterns were observed in scalp EEG recordings of epileptic patients[26], but these were without invasive recordings to serve as a ground truth. By contrast, in[27], Hofmeijer et al. monitored 18 stroke and 18 TBI patients using 21-electrode EEG systems, where visual inspection did not reveal any SDs. In[27], patients did not receive craniotomy, and there was no invasive recording. Therefore, it is not clear which patients had SDs. In a commentary paper[28] on this work, Hartings et al. provided potential reasons for the reported negative results in[27], including the lack of criteria for SD detection in scalp EEG, spatial low-pass filtering effect of the intact skull in these patients, and use of short one-hour time intervals for visual inspection. Using highly compressed EEG recordings in 11-hour time intervals, the authors in[28] reported visual identification of some SD depression patterns in a TBI patient, but without any invasive ground truth. This work was recently followed by[9], in which the authors monitored 15 TBI and 20 aneurysmal SAH (aSAH) patients, some of whom received craniotomy, with simultaneous invasive and noninvasive recordings. No pattern of SDs was found in continuous scalp EEG associated with invasively monitored SDs. Thus, it remains uncertain whether noninvasive detection of SDs is feasible, let alone sufficiently reliable for clinical relevance.

In addition, the feasibility of noninvasive SD detection has been explored using simulations[29–31]. While SDs are propagating depressions, in a related work on real data, we demonstrated the feasibility of localizing non-spreading depressions of activity (i.e., neural silences) in the brain using EEG recordings[32,33]—what we call Silence Localization. The insights obtained from Silence Localization[32,33] do not only suggest the possibility of detecting SDs using EEG, but also inform our approach in this work.

In this work, we explore the feasibility and quantify the performance of noninvasive SD detection in severe TBI patients using an automated algorithm applied to real data. We used our previous algorithm, WAVEFRONT[30], with appropriate modifications and improvements, for automated noninvasive detection of SD events in a group of 12 severe TBI patients who underwent decompressive hemicraniectomy (DHC) surgery,

followed by continuous monitoring in ICUs through simultaneous scalp EEG (a low-density EEG system with 19 electrodes at 10–20 standard locations) and intracranial ECoG recordings. A natural question is whether noninvasive and automated SD detection in patients with removed skull parts is clinically relevant? The DHC procedure is part of the standard of care for many severe TBI patients[34] to control elevated intracranial pressure, extract hematoma, and prevent further damage to the brain tissue[35–37]. It is worth noting that following a DHC, the scalp is sutured back over the brain, even though a piece of the skull is missing. Patients who receive DHC are continuously monitored in the ICU after their scalp incision is closed. During this period, while intracranial monitoring of SDs can provide higher spatial resolution[8], scalp EEG-based automated SD detection can provide valuable clinical information pertaining to worsening brain injury. Scalp EEG has broader spatial coverage than a locally placed ECoG strip. It also provides better spatial resolution in DHC patients compared to the intact skull EEG recordings[35], at least close to locations where the skull has been removed. Further, while procedural risks (e.g., bleeding, infection, etc.) associated with subdural electrodes are infrequent[38,39], noninvasive EEG precludes their possibility entirely. Therefore, noninvasive SD detection in severe TBI patients with DHC can prove clinically valuable in improving outcomes.

In Results, we show that WAVEFRONT achieved a reliable SD detection performance using Delta band EEG recording, with a ~ 74% average true positive rate (TPR) and less than 1.5% false positive rate (FPR) using cross-validation. Such a high TPR attained with a low FPR resolves the feasibility question: noninvasive SD detection is possible, at least for patients who have received DHC. However, is this performance sufficient for clinical goals? To answer this question, we performed an additional analysis, predicting the number of SDs from the total minutes of detected SD events. This analysis was inspired by Jewell et al.'s[10] estimation of SDs' frequency; unlike in our study, their aim was to automate *invasive* detection of SDs. Our preliminary results, albeit with limited data, suggest that WAVEFRONT can reliably estimate the number of SD occurrences in long 30-hour time intervals using a regression analysis. Overall, we believe that WAVEFRONT's performance indicates that noninvasive prognostication of worsening brain injury using SD detection is possible. However, to understand this potential, further studies with more data are warranted.

## Methods
In this section, we first describe the dataset used in our study, which included a group of 12 patients hospitalized after severe acute TBI who underwent DHC and cortical strip ECoG electrode placement. We then describe the intracranial SD ground truth labeling, and introduce our automated SD detection method, emphasizing the explainability of WAVEFRONT by providing intuition and visualization of the main steps.

**Dataset**. The dataset we used was obtained as part of a multi-center observational clinical study that monitored SDs in TBI patients (ClinicalTrials.gov ID: NCT00803036). Continuous EEG signals were recorded over a few days (95 ± 42.2-hour on average) following DHC using a DC-coupled EEG amplifier (CNS Advanced ICU EEG Amplifier from MOBERG ICU Solutions), with a sampling frequency of 256 Hz, from 19 electrodes placed at 10–20 standard locations.

In addition, during the DHC procedure, a strip of six monopolar ECoG electrodes (all the ECoG and EEG electrodes were referenced to a common electrode on scalp), with an interelectrode distance of 1cm, was placed on the hemisphere that

underwent the DHC, and continuous ECoG and EEG data were recorded simultaneously using the same amplifier. The recorded ECoG signals were visually assessed by a clinical expert (Dr. J. Hartings) to identify and annotate the SD episodes in the dataset. See the next section for a detailed description of SD temporal annotations.

For EEG preprocessing and artifact removal (see Supplementary Note 1 for details), we used two additional electrophysiological recordings: (i) an electrocardiogram (ECG) signal recorded at a 500 Hz sampling frequency and (ii) a plethysmogram (PLETH) signal at a 125$Hz$ sampling frequency. These signals were recorded using the IntelliVue Bedside (PHILIPS) patient monitor.

*Participants*. Data from 12 (nine male and three female) severe TBI patients were utilized in our study. Two patients had DHC in the left hemisphere, and the remaining 10 patients had DHC in the right hemisphere. Eleven patients experienced subdural hematoma (SDH), and one patient had an epidural hematoma (EDH). Detailed information about these patients is included in Table 1. All procedures were approved by the University of Cincinnati Institutional Review Board (Protocol ID 2016-8153). A legally authorized representative for each patient provided surrogate consent for participation in the initial research study. For visualization, computed tomography (CT) scans of a TBI patient (patient 6, see Table 1) with right DHC are shown in Fig. 1, where the locations of DHC and evacuation of hematoma can be seen as asymmetric dark regions in the skull's thick white layer (marked with green arrows), along with the scalp layer on the DHC site (marked with orange arrows). In addition, the location of the subdural strip is shown (Fig. 1 right).

**SD event temporal annotation based on ECoG signals**. Each SD event in these TBI patients was annotated over time by a clinical neuroscientist (Prof. Jed A. Hartings at Department of Neurosurgery, University of Cincinnati), through visual assessment of full-band ECoG signals. In this paper, an SD event refers to a unique SD wave, as annotated by consideration of all electrodes of the ECoG strip. Each unique SD wave was annotated at the start of a slow near-DC negative shift (in 1–10 mHz), i.e., slow potential change (SPC), in a chosen ECoG electrode, which is not always the same, even for the same patient. The temporal annotation of an SD event through visual inspection of the ECoG signals may not accurately reflect the actual onset of each SD wave. The reported performance metrics in this study (e.g., TPR and FPR) depended on the temporal annotations of SDs (see Results). Four different types of SD events were annotated: (i) CSD: an event during which there was a cortical spreading depression (CSD) in each electrode that had an SPC, where CSD was a manifestation of spreading depolarization and was defined as a cortical wave of depression in the high-frequency (> 0.5 Hz) ECoG (HF-ECoG) signals[40]; (ii) ISD: an event where the HF-ECoG signals at all the participating electrodes with SPC were already flat—these are called isoelectric spreading depolarizations (ISDs)[8]; (iii) CSD/ISD: an event where some ECoG electrodes experienced CSD-like propagation, while other electrodes had ISDs; and (iv) scCSD: an event which was identified as a clear SD in the signal of a single electrode. Although it appeared only on a single electrode, it met the consensus criteria, defined by Co-Operative Studies on Brain Injury Depolarizations (COSBID)[8], to be classified as an SD. According to COSBID, a minimal criterion to score an event as SD is "an event which has a characteristic DC shift associated with spreading depression of spontaneous activity even if DC shift and spreading depression are restricted to a single

**Table 1 Clinical details and demographic information for 12 TBI patients in the dataset.**

| Patient No | Age | Gender | Cause of Injury | GCS | Pupil Reactive | Lesions | Craniotomy Location | Bone Flap Replacement | Location of ECoG strip | EEG/ECoG Start (hours) | EEG/ECoG Duration (hours) | SDs in ECoG |
|---|---|---|---|---|---|---|---|---|---|---|---|---|
| 1 | 31 | M | Fall | 3 | 2 | R SDH | R | No | R Fr | 11 | 75 | 4 |
| 2 | 49 | M | Fall | 3 | 0 | L SDH | L | No | L Fr | 6 | 44 | 115 |
| 3 | 27 | M | MC | 5T | 2 | R Cont, SDH | R | No | R T | 7.5 | 113.5 | 128 |
| 4 | 23 | M | MVA-P | 2T | N/A | R A SDH | R | No | R T | 13.5 | 94.5 | 181 |
| 5 | 76 | F | Fall | 2T | 2 | R A SDH | R | No | R T | 21 | 7 | 71 |
| 6 | 71 | M | Fall | 14 | N/A | R SDH | R | No | R T | 14 | 84 | 10 |
| 7 | 71 | M | Fall | 7T | 2 | R SDH | R | No | R Fr | 8 | 87 | 0 |
| 8 | 60 | F | Fall | 14 | N/A | R SDH | R | No | R Pr | 20 | 70 | 10 |
| 9 | 72 | M | Fall | 5T | 2 | R SDH | R | No | R Pr | 30 | 133 | 57 |
| 10 | 49 | F | Fall | 6 | 2 | R SDH | R | No | R Fr | 17 | 289 | 5 |
| 11 | 30 | M | Fall | 9 | 2 | R EDH | R | No | R Fr | 14 | 13 | 0 |
| 12 | 23 | M | MVA | 3T | 2 | L SDH | L | No | L T | 8 | 128 | 119 |

Note: GCS and pupillary reactivity were assessed at admission to the study hospital following resuscitation. T is added to the GCS scores of intubated patients. For pupils, 1=one reactive, 2=both reactive, 0=neither. The number of annotated SD events (CSD, CSD/ISD, scCSD) based on the recorded ECoG dataset is given.

ECoG electrocorticography, EEG electroencephalography, GCS Glasgow Coma Scale, SD Spreading Depolarization, Cont contusion, EDH epidural hematoma, F female, Fr frontal, GOS Glasgow Outcome Score, L left, M male, MC motorcycle, MVA motor vehicle accident, MVA-P pedestrian involved in motor vehicle accident, N/A not applicable, R right, SDH subdural hematoma, T temporal, A acute, TBI traumatic brain injury, Veg vegetative state.

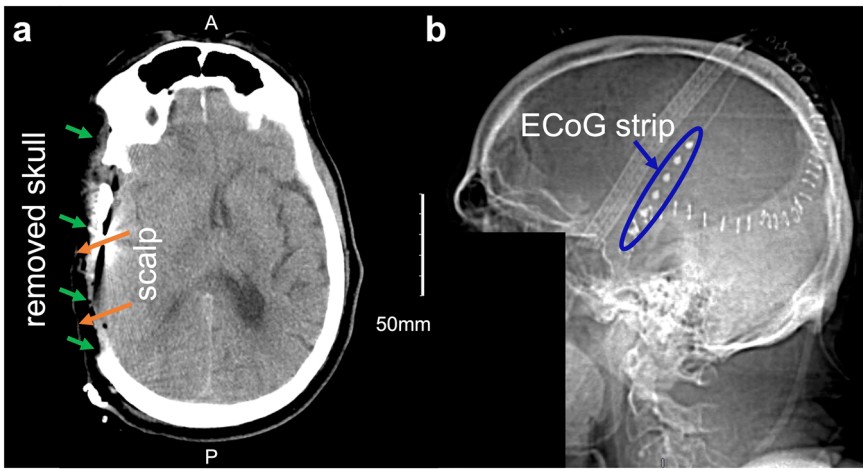

**Fig. 1 CT scan of a severe TBI patient with right DHC.** Transverse (**a**) and right side (**b**) view of computed tomography (CT) scans of a severe traumatic brain injury (TBI) patient (patient 6, see Table 1) with right decompressive hemicraniectomy (DHC). The missing portion of the skull (marked with green arrows), the scalp layer on the DHC site (marked with orange arrows), and the strip of six electrocorticography (ECoG) electrodes are also marked. The facial region is stripped away to ensure the anonymity of the patient.

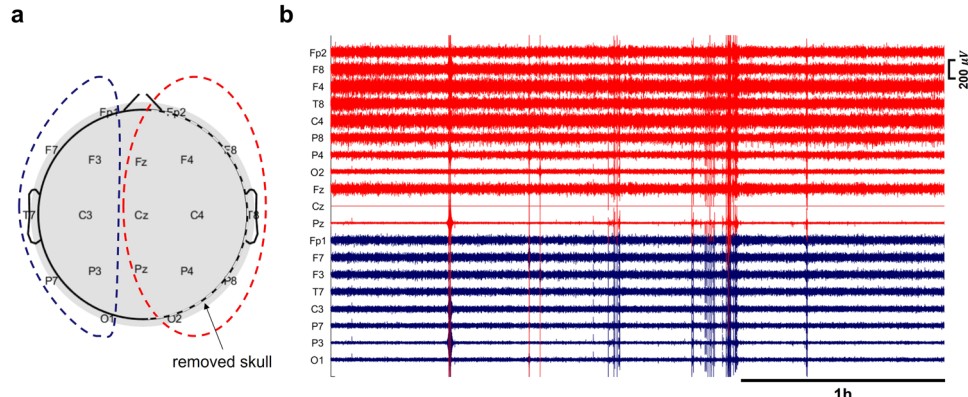

**Fig. 2 EEG baseline power in a patient with right DHC.** Electroencephalography (EEG) baseline power in ipsilateral (with missing skull) and contralateral (with intact skull) hemispheres in a patient (patient 4, see Table 1 for details) with right decompressive hemicraniectomy (DHC): **a** 11 ipsilateral EEG electrodes are marked with red dashed line, and eight contralateral electrodes are marked with blue dashed line and **b** 3 h of the EEG recording across ipsilateral (red traces) and contralateral (blue traces) electrodes. Most of the ipsilateral electrodes on top of the DHC region (with missing skull) had higher EEG baseline power (e.g., Fp2, F8, F4, T8, C4, and P8), in comparison with the contralateral electrodes on the regions with intact skull. The EEG signals were band-pass filtered in [0.5, 30] Hz and preprocessed (before amplitude outlier removal, see Supplementary Note 1 for more information). The signal at Cz had poor quality and was removed through the preprocessing steps.

channel." The total number of annotated SD events for each patient in the dataset is included in Table 1.

In this study, the ground truth SDs were annotated based on a single ECoG strip located in the DHC hemisphere (e.g., see Fig. 1), with no invasive measurements in the contralateral hemisphere (i.e., the hemisphere with an intact skull). We observed that the scalp EEG signals from the contralateral hemisphere and ipsilateral hemispheres were significantly different due to the missing part of the skull in the ipsilateral hemisphere (e.g., ipsilateral EEG signals had higher average power, see Fig. 2b and Supplementary Note 1 and Supplementary Fig. 1 for hemispheric comparison of average baseline power on the scalp, $p < 1e - 8$). The enhancement of EEG signals due to the defects in the skull is consistent with observations of "breach rhythm" reported in the literature, which is an increase in signal power in a wide range of frequencies in areas with skull defects[41,42]. In this paper, we only used ipsilateral scalp EEG electrodes in each patient for SD-related inferences. Excluding the contralateral electrodes is helpful to (i) tailor the WAVEFRONT algorithm to the SD events that we are certain about (i.e., events

that we have a ground truth for) during the training process and obtain a more realistic estimation of WAVEFRONT's performance in noninvasive detection of SDs, and (ii) acknowledge the statistical differences between the ipsilateral and contralateral EEG signals, as ignoring these differences may adversely affect the performance of WAVEFRONT. Because EEG signals tend to be less sensitive to contralateral sources, we expect this restriction to not hurt the performance of our algorithm. Finally, electrodes on the midline (Fz, Cz, and Pz) were included in our analysis, as they are sensitive to signals on either side. Figure 2a shows the selected subset of EEG electrodes (11 out of 19 electrodes) for a patient with a right hemisphere DHC.

**EEG preprocessing pipeline.** We used a multi-step preprocessing pipeline, described below, to prune the continuous EEG recordings and reject ICU-related artifacts and segments of EEG signals with poor quality electrode-scalp contacts:

*Band-pass filtering and downsampling.* We preprocessed the EEG data using the EEGLAB toolbox[43] in MATLAB. To be able to

evaluate the performance of our SD detection algorithm in different frequency bands, we bandpass filtered the EEG signals in different frequency ranges, namely, [0.001, 0.01]Hz (near-DC), [0.5, 4]Hz (Delta), [4, 8]Hz (Theta), [8, 12]Hz (Alpha), and [12, 30]Hz (Beta) using a Hamming-windowed sinc finite impulse response (FIR) filter. The filter order was 1000 with a transition bandwidth of ~ 0.02 Hz. An upper cutoff frequency of 30 Hz was used to remove high-frequency noise components. The filtered EEG signals were then downsampled to 64 Hz. We also bandpass filtered the ECG and PLETH signals in the frequency range of [0.5, 30]Hz and downsampled them to 64 Hz.

*Masking out poor-quality segments of EEG signal.* There are segments of the EEG recordings with poor quality electrode-scalp contacts which could be due to movements of patient on the bed, and conductive gel/saline drying out at each electrode, etc. CNS Advanced ICU EEG Amplifier monitors the quality of each electrode-scalp contact through continuous impedance recording at a sampling frequency of 1Hz. To enable our algorithm to detect SDs even when a few of the electrodes do not have good contact, we used these impedance recordings to implement masks for automated removal of the parts of EEG signals with poor quality EEG. We upsampled the continuous recording of impedance at each EEG electrode to match the sampling frequency of the EEG signals. For each electrode, the median of impedance over the whole recording was calculated ($M_{Imp}^{ch}$) and parts of the EEG signals with abnormally high ($>2M_{Imp}^{ch}$) impedance were used for masking. Instead of cutting these parts out of the EEG signals, we assigned dummy zero values to these parts (i.e., we mask out these parts) to maintain the continuity of the recordings. As it is explained in "Methods: SD detection and tracking using WAVEFRONT", these masked-out sections were excluded from the power envelope extraction and the downstream analyses to prevent false alarms resulting from these zero values. This helps us to keep the parts of the dataset where a few of the electrodes have good recording quality. In "Methods: SD detection and tracking using WAVEFRONT", we discuss in detail how we detected SDs when the recordings of some EEG electrodes were not available/usable. For the time intervals during which the signals of all EEG electrodes were masked out, dummy zero values are assigned to the PLETH and ECG signals as well for performing independent component analysis, discussed next.

*Artifact classification and removal using independent component analysis (ICA).* We grouped together the ECG, PLETH, and EEG signals and performed an ICA to extract and remove sources of artifact (such as eye blinks, eye movements, heartbeats, and muscle artifacts) in EEG signals. We used the EEGLAB[43] toolbox to calculate the independent components and used an automated EEG independent component classifier plugin (ICLabel) in EEGLAB to guide our decision on which components belong to sources of artifacts and subsequently removed them from the EEG recordings.

*Outlier removal.* Extracted ICA components may not perfectly separate some artifacts with abnormally high amplitudes (i.e., outliers) from the EEG signals. Therefore, as the last step in the preprocessing pipeline, we detected and removed the amplitude outliers using Tukey's fences method[44,45]. Tukey defines the outliers as data points that fall outside an interquartile range of $[Q_1 - k(Q_3 - Q_1),\ Q_3 + k(Q_3 - Q_1)]$, where $Q_1$ and $Q_3$ are the first and third quartiles respectively, and $k$ is an outlier threshold. We used $k = 3$, which detects far out data points according to Tukey's outlier definition. For each electrode, we detected and masked out (following the same procedure as impedance-based

masking discussed earlier in this section) parts of the EEG signals with outlier amplitude. Supplementary Fig. 2 illustrates the pre-processing steps applied on the ipsilateral EEG signals of a patient with right DHC, in a 4-hour time window. Supplementary Fig. 2a shows the full-band EEG signals, with poor quality (high impedance) in the first ~ 77 min of the recording, which is masked out in the band-pass filtered signals (in Delta band, see Supplementary Fig. 2b). The artifacts and outliers are detected and removed as it is shown in Supplementary Fig. 2c. The preprocessed continuous EEG signals were then used for SD detection using WAVEFRONT.

**SD detection and tracking using WAVEFRONT.** We used our previously proposed WAVEFRONT SD detection algorithm[30], with appropriate modifications and improvements, to detect and track SD waves in EEG recordings of 12 TBI patients in the dataset (see Table 1 for more details on these patients). WAVEFRONT is an explainable automated SD detection framework with intuitive steps and interpretable detection outputs and results. It addresses the challenges of noninvasive detection of SD waves in EEG (see Discussion for details) by breaking down the challenging task of detecting the whole propagating SD wave in the brain using noisy and blurry filtered scalp EEG signals into simpler tasks of detection and classification of disjoint SD wavefronts, following these steps: Power envelopes of the scalp EEG signals at each electrode are extracted, and depressions (power reductions) are detected. These detected depressions are then projected onto a 2D plane to obtain depression wavefronts. Propagating SD wavefronts are then detected and tracked based on their speed and direction of propagation. To estimate the speed and direction of propagation of depression wavefronts on these 2D planes, WAVEFRONT uses a computer vision technique called optical flow[46,47]. It then stitches together the detection of these wavefronts over time and space to detect and track SD waves in the brain. This overcomes the challenge related to the effects of sulci and gyri and enables the detection and tracking of complex patterns of propagation.

Although the simulation results of automated SD detection in ref. [30] are promising, we recognize that WAVEFRONT suffers from certain technical shortcomings and cannot be directly applied to real scalp EEG signals for SD detection: (i) it uses a fixed set of parameters (e.g., depression level/depth threshold, temporal score threshold, and spatiotemporal neighborhood radius); (ii) it is highly sensitive to the amplitude outliers; (iii) it implicitly assumes the power level of normal background brain activity (DC offset of the power envelope) is stable and not changing over time (see Figs. 9 and 10 in ref. [30] for more details), which limits the ability of WAVEFRONT in the detection of depressions, as well as near-DC shifts during propagating SDs in the real EEG recordings; (iv) it does not address the challenges of using a low-density EEG grid, including the high rate of false alarms due to the non-propagating depressions on the scalp that we observe here; (v) it does not address the challenges of using a very low-density EEG grid, including the high rate of false alarms due to the non-propagating depressions on the scalp; and (vi) it estimates the optical flows in pixels on the 2D images, rather than in terms of the physical distances on the scalp, which can introduce errors in estimation of the speed and direction of propagation of SD wavefronts.

In this work, we addressed these limitations of WAVEFRONT by making necessary modifications and improvements, including designing a training and validation framework for the algorithm to learn an optimal set of parameters through a cross-validation analysis. Other modifications include (a) designing a rigorous and automated preprocessing pipeline for outlier rejection and

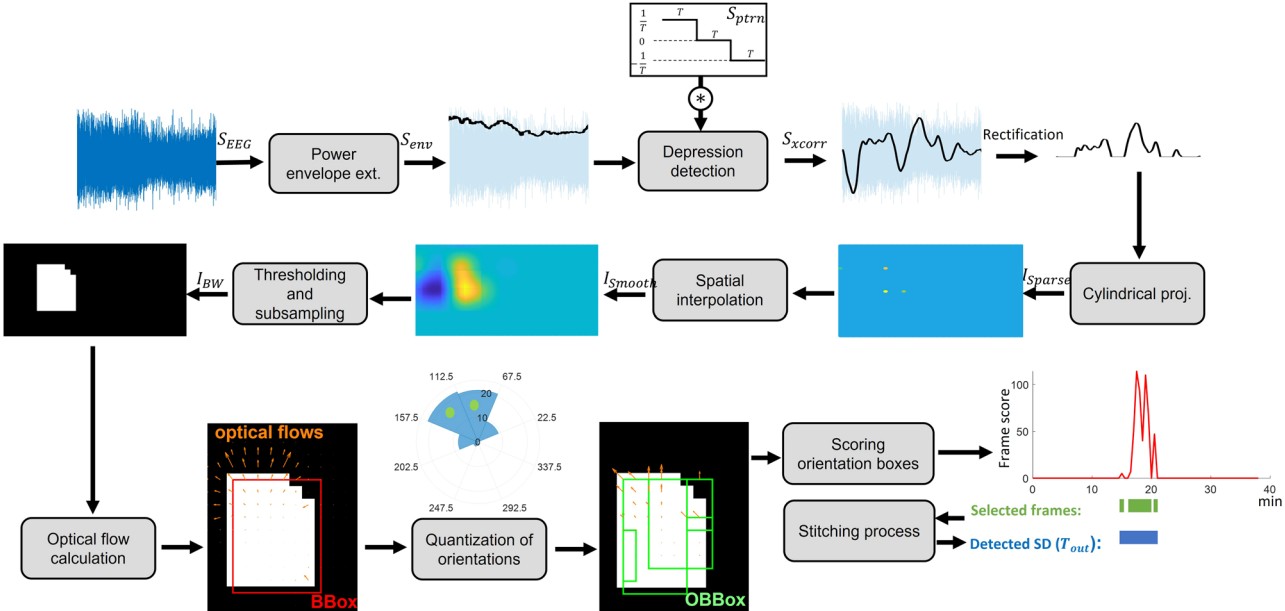

**Fig. 3 Main steps of the WAVEFRONT algorithm.** The power envelope of the preprocessed electroencephalography (EEG) signals ($S_{EEG}$) was extracted, and cross-correlated with a first-derivative kernel ($S_{Ptrn}$) to extract power depressions as large positive peaks in $S_{Xcorr}$, which were rectified and projected on a 2D plane through a cylindrical projection. The resulting image ($I_{Sparse}$) was then spatially interpolated, thresholded, and subsampled to obtain binary images ($I_{BW}$), where the depression wavefronts were captured as white contiguous pixels. The movement of wavefronts in these binary images was estimated using optical flows. Dominant directions of propagation (marked bins in the orientation histogram) were found through quantization of orientations and scored based on the consistency and speed of propagation of wavefronts. Candidate frames were selected based on the calculated scores and stitched together for the final detection output ($T_{out}$).

pruning the EEG signals, (b) using a power-envelope extraction method that is less sensitive to large-amplitude artifacts (i.e., outliers), (c) extending the depression extraction method to be able to detect DC shifts in the near-DC components (1–10 mHz) as well as the power depressions in the higher frequency bands ($\geq 0.5$ Hz), (d) defining an effective propagation measure along with a learnable threshold on this measure to reject the false alarms of the non-propagating depressions on the scalp, and (e) mapping the estimated optical flows on the scalp spherical surface. Following are the steps of the WAVEFRONT algorithm, along with the details on the modifications and improvements we make in each step (see Fig. 3):

*Epoching and envelope extraction.* We extracted epochs from the preprocessed EEG signals using overlapping time windows of 240 min with a step size of 180 min. For each epoch, the EEG signal at each electrode ($S_{EEG}$) was normalized by its estimated standard deviation. As discussed in Methods, we only used ipsilateral scalp EEG electrodes for each patient because of the missing spatial SD ground truth in the contralateral hemisphere, and heterogeneity of the baseline EEG power between the hemispheres with DHC and the hemisphere with an intact skull. Based on the results, for all of the 12 patients in the dataset, there is a statistical difference ($p < 1e - 8$) between the ipsilateral and contralateral average power (see Supplementary Note 1 for hemispheric comparison of average baseline power on the scalp). However, even in the hemisphere with DHC, the scalp electrodes which are far from the site of surgery have overall lower baseline power, in comparison to the electrodes which are placed right on top of the regions with a missing skull (see Fig. 2b as an example, where P4, O2, and Pz have smaller baseline power in comparison with the rest of ipsilateral electrodes shown in red). This epoching and power normalization step addresses the heterogeneity of the baseline EEG power across electrodes and over time and helps to detect and extract the depressions in electrodes with low baseline

power, which are otherwise missed in the interpolation and thresholding step (this step is explained later in this section). In addition to the baseline power normalization, this epoching helps in the parallelization of the downstream data analysis and the training process explained in Results.

Following the normalization step, the amplitude values were squared, and upper root-mean-square (RMS) envelopes of the power signals were extracted using a sliding time window of 5 min. We used the implementation of the RMS envelope extraction method in ref. [48]. There might be some small and isolated valid portions of the EEG signals at each electrode with the normal recording quality, which were interleaved by dummy zero values following the masking out step in "Methods: EEG preprocessing pipeline". These portions were not large enough to capture the slow depressions of SDs across EEG electrodes. To prevent false alarms resulting from these isolated intervals, at each epoch and before power envelope ($S_{env}$) extraction, we masked out small intervals of the EEG signals which were less than 20 min long, and isolated, i.e., more than 1 min apart from the nearest valid intervals. The masked-out sections of the EEG signals were excluded from the envelope extraction.

*Power depression extraction.* In this step, we detected and extracted the power depressions at each electrode based on the power envelopes ($S_{env}$). In order to detect the falling edges of the power envelopes, which are followed by a prolonged power reduction, we cross-correlated $S_{env}$ with a piecewise-constant function as a first-order derivative kernel (see $S_{Ptrn}$ kernel in Fig. 3). This kernel extracts EEG power depressions as large positive values in the cross-correlated signal ($S_{Xcorr}$). The 5 min width used for envelope extraction and depression edge detection is small in comparison to the large temporal width of depressions in severe TBIs. The same first-order derivative kernel was used for the detection of DC shifts in the near-DC components ([0.001, 0.01] Hz). This kernel was directly applied on $S_{EEG}$, after the

epoching and power normalization step, to detect the falling edges of SPCs in the near-DC components. Due to the low density of EEG electrodes on the scalp (only 19 electrodes), Laplacian spatial filtering[49,50] is not effective to extract narrow SDs[30,49], and hence we did not use it in this study.

*Cylindrical projection.* We closely followed the steps in ref. [30] to project the extracted $S_{Xcorr}$ signals at ipsilateral EEG electrode locations on a 2D plane. Before this projection, $S_{Xcorr}$ signals were rectified, i.e., negative values (corresponding to the rising edges in the power envelopes) were zeroed, and positive values, which correspond to the falling edges of the depressions, were kept unchanged. Figure [3] shows an example of the resulting image ($I_{Sparse}$) using this projection for patient 3, around an annotated SD event. The corresponding scalp electrode locations in these 2D plane are shown in Supplementary Fig. 3 for patients with left and right DHC. At each time point, the median value of the $S_{Xcorr}$ signals at ipsilateral electrode locations was assigned to the rest of the pixels in $I_{Sparse}$.

*Interpolation and thresholding.* We spatially interpolated $I_{Sparse}$ using a 2D Gaussian kernel with $\sigma = 2.62$ cm. This large standard deviation was chosen because of the low density of EEG electrodes, where the average inter-electrode distance is ~ 5.4 cm. For interpolation at the boundaries, each $I_{Sparse}$ image was padded with the median of the corresponding ipsilateral values of the $S_{Xcorr}$ signals. The resulting smooth image ($I_{Smooth}$) is shown in Fig. [3]. Following this step, we introduced a thresholding mechanism to extract binary images from the smooth 2D images ($I_{Smooth}$): (i) we assigned zeros to the pixels at each image whose values were $\leq M_{I_{Smooth}} + Thr_1(H_{I_{Smooth}} - M_{I_{Smooth}})$, where $M_{I_{Smooth}}$ and $H_{I_{Smooth}}$ are the median and the maximum of the pixel values at each $I_{Smooth}$ image respectively, and $Thr_1$ is the depression-level threshold at each time point which were found through the training process as a learnable parameter (see Results for details on the training and validation steps). This step zeros the pixels with values close to or less than the median value at each image, and set the pixels with large positive values in $I_{Smooth}$ as candidates for SD wavefronts, (ii) In addition, we assigned zeros to the whole images where most of the EEG electrodes were masked out (images with less than 5 participating scalp electrodes out of the total 11 ipsilateral electrodes). This rejects the binary images with poor EEG signals, (iii) we set the remaining pixels (assign 1's), and finally rejected (assign zeros) the images where more than half of the pixels are set. This was done with the reasonable assumption that SD depressions cannot spatially expand over more than half of the cortical surface. Following this three-stage thresholding mechanism, binary images ($I_{BW}$ in Fig. [3]) were extracted, where non-zero pixels form connected components which are either parts of the SD wavefronts or non-SD activities in the brain. Through the following steps, we classified these connected components and detected and tracked SD wavefronts.

*Subsampling and optical flow calculation.* We temporally subsampled the series of binary images ($I_{BW}$) every 30 s. Since the extracted power envelopes and depression signals ($S_{Xcorr}$) have a very slow temporal pattern in the order of minutes, this temporal subsampling significantly reduces the computational complexity of the WAVEFRONT for noninvasive detection of SDs in these TBI patients, without adversely affecting the SD detection performance. In addition, due to the very low density of the CNS EEG grid in this study (only 19 electrodes), we spatially subsampled $I_{BW}$ images so that the inter-electrode distances in these 2D images are less than three pixels. This helps to better capture the propagation of SD wavefronts across EEG electrodes in these

binary images and further reduces the computational power required for the downstream analysis in WAVEFRONT. We used the bicubic interpolation method[51], and its Matlab implementation[52], for spatial subsampling of $I_{BW}$. We carefully chose the parameters of the interpolation kernel (standard deviation $\sigma$) and the spatial subsampling rate so that the corresponding pixels of each electrode location on the scalp have a representation in the lower resolution binary image. Therefore, we do not expect to have missed any connected component (wavefront) in the lower resolution binary image. After these subsampling steps, we closely followed the steps in ref. [30] for the calculation of optical flows to capture the movements of connected components and SD wavefronts.

Advancing on[30], we made minor modifications and improvements to the way we estimate optical flows. Optical flow is a computer vision technique to track moving objects across frames of a video[46,47]. It uses the spatiotemporal brightness variations of the pixels to estimate the velocity (magnitude and direction) of the moving objects. The optical flows of the depressions were calculated based on the 2D images, which are the cylindrical projections of scalp electrode locations onto a 2D plane. In these 2D images, the horizontal dimension captures the azimuthal angle ($\phi$), and the vertical dimension captures the polar angle ($\theta$, with $\theta = \frac{\pi}{2}$ to be at the north pole) of the electrode locations in the spherical coordinate. Therefore, to estimate the optical flows of the connected components (the contiguous non-zero pixels in $I_{BW}$) in spherical coordinates, based on the calculated optical flows in the 2D images, we defined the following mappings for the vertical and horizontal optical flow magnitudes:

$$\begin{aligned} V_x(\phi_x, \theta_y) &= r\Delta\phi V_x^{2D}(x,y)cos(\theta_y) \\ V_y(\phi_x, \theta_y) &= r\Delta\theta V_y^{2D}(x,y) \end{aligned} \tag{1}$$

where $r$ is an average human head radius (we used $r = 75$ mm in this study), $V_x^{2D}(x,y)$ and $V_y^{2D}(x,y)$ are the horizontal and vertical magnitudes of the estimated optical flow at $(x,y)$ on the 2D plane, and $V_x(\phi_x, \theta_y)$ and $V_y(\phi_x, \theta_y)$ are their corresponding magnitudes on the spherical model of the scalp. $(\phi_x, \theta_y)$ is the corresponding spherical coordinate of the $(x,y)$ location in the 2D plane, and $\Delta\phi$ and $\Delta\theta$ are the azimuthal and polar resolution of each pixel in the 2D images. Supplementary Fig. 4 shows an example of this mapping for an optical flow. We used the mapped $\mathbf{V} = (V_x, V_y)$ spherical optical flows instead of the original 2D optical flows $\mathbf{V}^{2D} = (V_x^{2D}, V_y^{2D})$ throughout the next steps of WAVEFRONT. It is worth mentioning that $\mathbf{V} = (V_x, V_y)$ is an estimation of the extent of movements for the connected components in *mm* on the spherical scalp surface rather than in pixels on the 2D images. This makes it easier to impose constraints on the speed of propagation of connected components for the detection of SD wavefronts (see the "Scoring OBBoxes based on the consistency of propagation" subsection for more details).

*Quantization of orientations.* We closely followed the steps in our previous work[30] to assign bounding boxes—i.e. BBox—to the connected components in $I_{BW}$, calculate an orientation histogram for each BBox and quantize the orientations of optical flows, and finally extract prominent direction(s) of propagation in each BBox. In this study, we introduced an additional constraint on the effective propagation of each BBox before quantization. This additional step was designed to reject the pop-up/fade types of transition of the connected components in the binary images around each electrode location in $I_{BW}$. Supplementary Fig. 5a shows an example of pop-up/fade transition in the binary images, while Supplementary Fig. 5b shows a BBox with significant effective propagation. The low density of the EEG grid in this

study makes it impossible to capture small movements of SD wavefronts unless they propagate sufficiently across the 2D planes. Therefore, to reduce the false alarms because of these pop-up/fade transitions, and only considering the propagating depressions across scalp electrodes, we tried to detect non-propagating BBoxes and remove them. We defined and calculated the effective propagation measure or EPM for each BBox, based on the estimated optical flows, as follows:

$$\text{EPM} = \sqrt{\left(\frac{1}{N_{opt}}\sum_{i=1}^{N_{opt}}\|\mathbf{V}_i\|\,cos(\alpha_i)\right)^2 + \left(\frac{1}{N_{opt}}\sum_{i=1}^{N_{opt}}\|\mathbf{V}_i\|sin(\alpha_i)\right)^2}$$

$$(2)$$

where, $\|\mathbf{V}_i\|$ and $\alpha_i$ are the magnitude and orientation of the $i^{th}$ optical flow in the BBox, and $N_{opt}$ is the total number of optical flows in that BBox. The first term under the square root in (2) captures the average horizontal magnitude, and the second term captures the average vertical magnitude of the flows in a given BBox. EPM can take values between 0 and 1, where $EPM = 0$ indicates that all optical flows in the BBox are directed either inward (a fading connected component), outward (a popping-up connected component), or have zero magnitudes (no movement in the corresponding connected component). We apply a threshold on EPM values of BBoxes and remove the BBoxes and their optical flows if $EPM < Thr_2$, where $Thr_2$ is a learnable parameter in the modified WAVEFRONT algorithm (see Results for details on the training and validation process). The remaining optical flows were quantized using the orientation histograms, following the steps in[30]. Due to the low-resolution EEG grid in this study, we used a coarser orientation histogram with 8 bins of 45° each.

*Orientations bounding boxes (OBBox).* We extracted the orientation bounding boxes (OBBox) using the quantized orientations of optical flows (green boxes in Fig. 3), closely following the steps in ref. [30].

*Scoring OBBoxes based on the consistency of propagation.* We closely followed the steps in our previous work[30] to score the OBBoxes. We used a spatial radius of 7cm (to cover the large inter-electrode distances in this low-resolution EEG grid), and a temporal range of $Thr_3$ min ($Thr_3$ min before and $Thr_3$ min after the current frame) to find the neighbors of each OBBox. The algorithm learns the temporal range of $Thr_3$ through the training process. We imposed speed constraints on the propagating wavefronts and removed the OBBoxes with very fast (> 8 mm/min) or very slow (< 0.5 mm/min) propagation, counted the number of matching OBBoxes for each of the remaining boxes, and considered this count as a spatiotemporal score to each OBBox. In addition, we calculated the temporal score for each OBBox to measure the consistency in the propagation of wavefronts over time. If the fraction of the number of frames with non-zero temporal scores over the total number of frames in the $Thr_3$ min neighborhood is less than $Thr_4$ (i.e., only a small number of frames contribute to the spatiotemporal score), we remove the corresponding OBBox (i.e., assign zero to its spatiotemporal score). $Thr_4$ is another learnable parameter in the WAVEFRONT algorithm and takes values between 0 (no contributing frame to the score of the OBBox) and 1 (all frames in the temporal neighborhood of the current frame, defined by $Thr_3$, contribute to the score of the corresponding OBBox).

*Stitching process and the final decision on detection.* We rejected the OBBoxed with small spatiotemporal scores (less than 1% of the maximum available score at each frame), and rejected the

frames with small frame scores (scores of less than 5% of the median of the frame scores). As the final step, we stitched together the selected frames using a sliding time window of 2min and closely followed the steps in ref. [30] to obtain the final temporal detection output $T_{out}$ (1=detected SD at the corresponding frame, 0=no SD wavefront was detected at the corresponding frame). Supplementary Fig. 6 shows an example of SD detection for patient 6, with a single isolated SD event in a ~ 3-hour time window. Please note that in this study, we did not use the spatial detection output ($I_{out}$) of the WAVEFRONT algorithm for performance evaluation since we lacked the spatial ground for SD wavefronts. Detailed discussions on the limitations of WAVEFRONT and the ground truth annotations are included in Discussion.

**Reporting summary**. Further information on research design is available in the Nature Portfolio Reporting Summary linked to this article.

## Results
In this section, we quantify the performance of our modified WAVEFRONT algorithm, as described in Methods, on 12 TBI patients in the SD-II dataset (see Table 1 and Methods for more details on these patients). We explore the generalizability of our algorithm through a cross-validation analysis and compare the performance of WAVEFRONT across different EEG frequency bands. With an emphasis on the trustworthiness of our method, we provide different visual illustrations of the SD detection results, including temporal and spatial visualization of representative SD events, and carefully define the performance metrics we used. Finally, we evaluate the performance of WAVEFRONT in measuring the frequency of SDs (the number of occurrences) in large (30-hour) time windows using regression analysis.

**Performance rules and metrics**. We assessed the average SD detection performance of WAVEFRONT by examining the performance on overlapping time windows, each with a width of $WL = 2$min and step size of $\Delta W = 30$ s. In doing so, we used the following conditions and definitions:

If a time window includes an annotated SD, it is called an SD window. An SD window is said to be detected by WAVEFRONT if there exists a non-zero $T_{out}$ value within a temporal distance of $\Delta t$ from the annotated SD in the SD window. We chose $\Delta t = 1$-hour, because of the following reasons: (i) for visually observed SDs in DHC patients based on noninvasive EEG signals, the reported time interval between the lowest depression points at two scalp electrodes is 17 min (median) with 11–34 min interquartile range[24]; therefore, we examined within $\Delta t = 1$h around each SD annotated event; (ii) the average interelectrode distance of the EEG system used in this study is ~ 5.4 cm, and the reported range of speed of SD propagation is 1–8 mm/min[40]; consequently, it takes ~ 54 min for the slowest depression to propagates between each pair of electrodes on the scalp; (iii) in addition, the temporal annotation of SD was extracted using the ECoG strip which only covers a local region, while the detection is based on the ipsilateral EEG electrodes which cover the whole DHC hemisphere. Therefore, an SD event have been detected at any time within the duration of propagation, and hence within a temporal distance from its annotation. This is a limitation of the SD ground truth in this study, and the choice of $\Delta t = 1hr$ is only an assumption, which may have resulted in slight over or underestimation of the actual performance of WAVEFRONT in the detection of SD events. A performance metric of the true positive rate (TPR) was defined based on the SD windows as

follows:

$$\text{TPR} = \frac{\text{number of detected SD windows}}{\text{total number of SD windows}} \quad (3)$$

If a time window includes detection (intervals with non-zero $T_{out}$), and no annotated SD is found within a temporal distance of $\Delta t$ from the detected intervals in that window, it is considered a false alarm window. In addition, time windows without any detection and any annotated SD inside or within a temporal distance of $\Delta t$ from either end of the windows are considered true negative windows. A performance metric of the false positive rate (FPR) is defined based on the true negative and false alarm windows as follows:

$$\text{FPR} = \frac{\text{number of false alarm windows}}{\text{number of false alarm windows} + \text{number of true negative windows}} \quad (4)$$

For diagnosis of worsening brain injury and clinical applications, achieving a low false alarm rate is crucial to minimize the risks and side effects of unnecessary treatments and interventions, especially invasive interventions such as DHC, for minimizing secondary brain injury. However, since the dataset was highly imbalanced (the number of non-SD windows was much larger than the number of SD windows), a seemingly low FPR could still have had a large number of false alarm windows. Therefore, we used an additional performance metric, called precision or positive predictive value (PPV)[53], defined as:

$$\text{PPV} = \frac{\text{number of detected SD windows}}{\text{number of detected SD windows} + \text{number of false alarm windows}} \quad (5)$$

We also defined $Q_{avg}$, a measure of signal quality for each (sliding, 2 min) time window, as the number of electrodes, averaged over the 2-min interval, that are not masked out in the window over the hemisphere with DHC. There are 11 electrodes ipsilateral to the site of ECoG placement (see Fig. 2a) since we included electrodes on the midline in the ipsilateral set. We chose a threshold of $Q_{avg} \geq 6$ for defining whether a time window has good-quality recordings. Thus, time windows with $Q_{avg} < 6$ were excluded from the SD detection performance calculations. In all, there were 36,709 excluded poor-quality windows (approximately 28% of the windows) across 12 patients. This large number of poor-quality windows was mainly due to the long time intervals during which the patients were disconnected from the EEG amplifier for procedures or imaging. During these intervals, the recordings were not stopped. Other poor-quality intervals may have been in partly due to the inherent limitations of scalp EEG recordings, e.g., low density of EEG electrodes (only 11 ipsilateral) at ICUs increases the chance of recording intervals with almost no reliable EEG signal. Higher spatial coverage of EEG electrodes on the ipsilateral hemisphere can mitigate this issue. In addition, DHC patients have highly concave scalp surface on the hemisphere with missing skull[35], making the scalp electrode placement and creation of good electrode-scalp contact even more challenging. This was another important contributing factor for the large number of poor-quality windows in this study. Nevertheless, there was a large number of remaining good-quality SD and non-SD windows (92,583 in total), which were distributed across the 12 patients and used for the calculation of WAVEFRONT's SD detection performance.

All bounds on the average reported TPR and FPR performance metrics reported here are 95% confidence intervals, which are estimated using the weighted bootstrapping method[54,55], with a bootstrap sample size of 100 (randomly selected with replacement). The weights are the number of non-SD windows for FPR confidence intervals and the number of SD windows for TPR confidence intervals.

## Testing WAVEFRONT's generalizability using a cross-validation analysis

*Leave-2-out cross-validation.* To evaluate the generalizability of WAVEFRONT and detect and prevent overfitting of our algorithm[56], we used cross-validation. We split the dataset into sets of train and validation patient groups, found the optimal sets of parameters for WAVEFRONT on the train sets, assessed the SD detection performance on the validation sets, and averaged the performance on different validation sets. Specifically, we used Leave-2-out cross-validation (L2O CV). We chose two patients out of the total 12 patients and left them out for validation in $\binom{12}{2} = 66$ different ways. For each of these 66 choices, we finetuned WAVEFRONT parameters using the 10-patient training set, following the steps below:

Using the defined performance metrics in the previous section, we optimized the parameters $(Thr_1^{opt}, Thr_2^{opt}, Thr_3^{opt}, Thr_4^{opt})$ in WAVEFRONT for the best possible train performance (i.e., lowest possible FPR) while maintaining a good detection power (high TPR) and precision (high PPV). This optimized set of parameters was then used to evaluate WAVEFRONT's performance on the corresponding validation set. We used a brute-force grid search to find the best set of parameters (list of search grids is included in Supplementary Table I). The following are the training and validation steps for each pair of train-test sets: (i) using the values in the search grids and for each set of $(Thr_1, Thr_2, Thr_3, Thr_4)$, the performance of WAVEFRONT was evaluated on the train set; (ii) a TPR threshold ($Thr_{TPR}$) was then applied on the train performance, and among the sets of parameters with train TPR $\geq Thr_{TPR}$, the one with the minimum FPR was chosen as the optimized set; (iii) WAVEFRONT, using the optimized set of parameters, was applied on the validation set to obtain the validation performance. We repeated these steps for all of the train-validation pairs and averaged the train and validation performance across all of the 66 pairs of train-validation sets. We calculated the average performance for different $Thr_{TPR}$ values in the range of [0.5, 0.85]; and (iv) the optimal operating point (i.e., $Thr_{TPR}^{opt}$) and its corresponding optimal set of parameters were found using the underfitting-overfitting (also known as bias-variance) tradeoff[57]. We define a cross-validation root-mean-square error (RMSE) using the averaged TPR and PPV values as $\epsilon_{CV} = \sqrt{(1 - \text{TPR})^2 + (1 - \text{PPV})^2}$. The average validation performance corresponding to the minimum cross-validation error $\epsilon_{CV}$ with PPV $\geq 0.5$ was reported as the generalization performance. PPV $= 0.5$ is the threshold at which only half of the detected intervals are true positives and the other half are false positives.

The TPR-FPR tradeoff was captured in a receiver operating characteristic (ROC) curve. We closely followed the threshold averaging method in[58] to generate the average train and validation curves. Figure 4a shows the average train (solid blue line) and validation (solid red line) ROC curves of WAVE-FRONT's performance in the detection of SDs using scalp EEG Delta band ([0.5, 4]Hz) with TPR values along the vertical axis and FPR values along the horizontal axis. PPV line of PPV $= 0.66$ is overlaid on top of the ROC curves, where the PPV $> 0.66$ region is located above the corresponding PPV line. Figure 4b shows the cross-validation error ($\epsilon_{CV}$), color-coded across different points in the validation ROC curve, where the point with the minimum error (i.e., optimal operating point) is marked. Based on the results, using Delta frequency band scalp EEG, WAVEFRONT achieves an average validation performance of TPR $= 0.74 \pm 0.03$

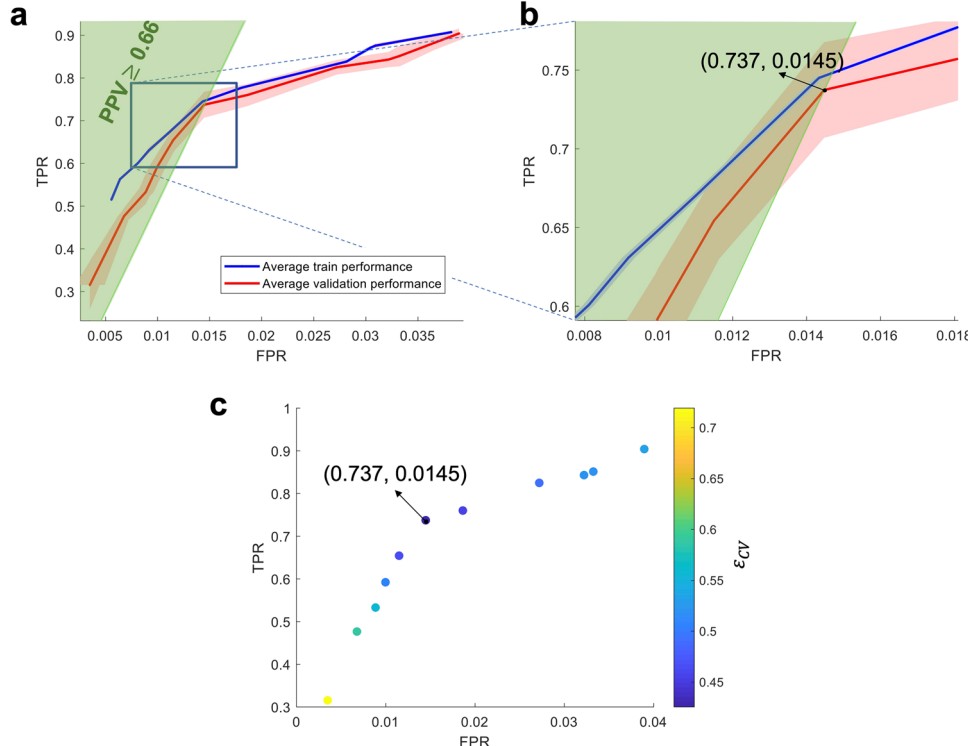

**Fig. 4 WAVEFRONT performance evaluation using a cross-validation analysis. a** Receiver operating characteristic (ROC) curves of the average train (solid blue curve) and validation (solid red curve) performance of WAVEFRONT in the detection of spreading depolarization (SD) events, using noninvasive scalp electroencephalography (EEG) signals in the Delta frequency band. The shaded blue and red regions show the 95% confidence intervals for the average train and validation curves, respectively. The vertical axis indicates true positive rates (TPRs), and the horizontal axis shows false positive rates (FPRs). The green highlighted region indicates the positive predictive values of $\geq 0.66$ (PPV), **b** Zoomed-in version of the ROC curves around the optimal validation operating point (TPR = 0.74 ± 0.03, and FPR = 0.0145 ± 7.57 × $10^{-4}$; in 95% confidence intervals), **c** Cross-validation error ($\epsilon_{CV}$), color-coded across different points in the validation ROC curve, where the point with the minimum error (i.e., optimal operating point) is marked.

(12,303 of 16,685 total SD windows were detected) in the PPV > 0.66 ROC region, with FPR < 0.015 (0.0145 ± 7.57 × $10^{-4}$, 6339 of 437,849 total non-SD windows were falsely detected). All the reported results are in 95% confidence intervals. This operating point in the average ROC curve corresponds to an optimal set of parameters as $Thr_1^{opt} = 0.3$, $Thr_2^{opt} = 0.6$, $Thr_3^{opt} = 0.69$, and $Thr_4^{opt} = 2$. This point has the smallest cross-validation error of 0.4256, while the points with higher TPR show an increasing trend in the cross-validation error (i.e., overfitting[56,57]), and points with lower TPR have larger error as well (i.e., underfitting[56,57]). In this study, overfitting is inevitable due to the small number of patients. We expect WAVEFRONT to achieve a better validation performance using a larger dataset for the training process. This requires further investigation when we receive access to the recordings of more patients with SDs.

We estimated the speed of propagation of detected SD events at the found optimal performance point in the Delta band. For each true detection event (connected 1's in $T_{out}$ which laid within $\Delta t$ temporal distance of the annotated SD events), we averaged over the magnitude of optical flows of the corresponding detected OBBoxe in the scalp spherical coordinates ($\|V\| = \sqrt{V_x^2 + V_y^2}$, see Methods for more details). Supplementary Fig. 7 shows the histogram of the estimated speed of propagation for the true detection events in the Delta band. Although no ground truth is available for the speed of propagation of annotated SD events in the dataset, this helps in understanding and comparing the range of SD speeds in the scalp EEG with the available scientific literature. Based on the results, the estimated speed ranges from 0.9 to 6.8 mm/min, which is a slightly smaller range in

comparison to the imposed speed constraints in WAVEFRONT ([0.5, 8] mm/min, see Methods), with the largest population around 3.6 mm/min. This observation is consistent with the widely reported range of 1–8 mm/min in the literature.

Figure 5 includes a spatiotemporal visualization of a sample SD propagation event in patient 4 with clustered SDs. We ordered the $S_{Xcorr}$ time series in Fig. 5b using transverse and longitudinal montages of the EEG electrodes (see Fig. 5a) to make it easier to visually track the propagation of depressions in the extracted $S_{Xcorr}$ signals (see Methods). In this example, the SD wavefront starts at Fp2 and Fz and gradually travels towards F4, ending at T8 and C4 (spatial propagation of this SD wavefront is shown in Fig. 5d). Figure 5c shows the MRI (left) and CT (right) scans of this patient, where the location of the six ECoG electrodes (right frontotemporal) is shown along with the locations of lesions and DHC. In this sample visualization in Fig. 5, there are eight clustered SD events (more than two SDs in a time interval of three hours or less[10,19]), where WAVEFRONT detects five SD events in three detection intervals (blue strips in Fig. 5b). This illustrates how WAVEFRONT, at least with low-density EEG, can under-detect SD events, especially when they are clustered.

In Supplementary Fig. 6, a similar visualization is shown for a single and isolated SD event in patient 6 (see the CT scan of this patient in Fig. 1). In this example, the SD depression first shows up at Fp2 on the scalp and slowly propagates towards F4, ending at C4 (a longitudinal path); at the same time, another propagation is observed far away from the ECoG strip (shown as six small red circles), following the path of P4-Pz-O2. The observed depression at C4 is spatiotemporally consistent with the intracranially annotated depression at $t_3$.

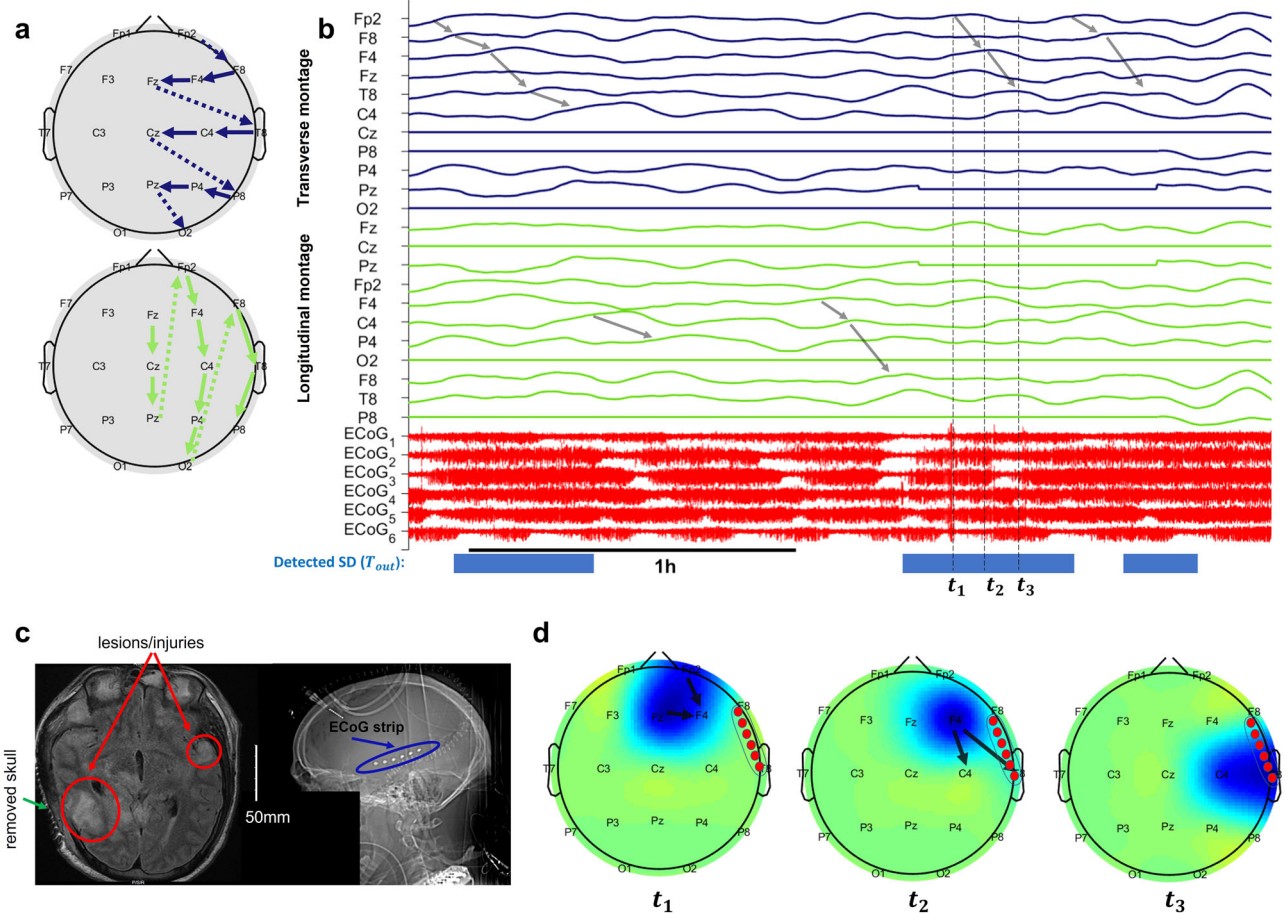

**Fig. 5 Visualization of a sample SD event in patient 4 with clustered SD events. a** transverse and longitudinal montages of ipsilateral electroencephalography (EEG) electrodes. These montages order the electrodes so the signals of anatomically neighboring electrodes are located next to each other in the temporal plots. **b** Time traces of $S_{Xcorr}$ and electrocorticography (ECoG) signals, where three time-points of the selected spreading depolarization (SD) event are marked as $t_1$, $t_2$, and $t_3$ with maximum depressions (peak in $S_{Xcorr}$) at (Fp2, Fz), F4, and (C4, T8) respectively. **c** Magnetic resonance imaging (MRI, on the left) and computed tomography (CT, on the right) scans of this patient, where the locations of lesions and injuries are shown, along with the right decompressive hemicraniectomy (DHC) region and the intracranial strip of ECoG electrodes. **d** Scalp topography of SD depressions at the three corresponding timepoints. The intracranial ECoG strip is located around the right frontotemporal lobe. The detected events ($T_{out} = 1$) using WAVEFRONT are marked with blue strips in (**b**), where the under-detection of WAVEFRONT is apparent, along with some missed detection intervals. A total of five SD events are detected in EEG; eight are marked in the ECoG signals. Some of the propagating depressions in (**b**) are marked with gray arrows across the $S_{Xcorr}$ signals, which correspond to the SD events in this time window.

## SD detection performance using different scalp EEG frequency bands.

We band-pass filtered the scalp EEG signals in different frequency ranges—[0.001, 0.01]Hz (near-DC), [0.5, 4]Hz (Delta), [4, 8]Hz (Theta), [8, 12]Hz (Alpha), and [12, 30]Hz (Beta)—to explore the feasibility and performance of noninvasive detection of SD events across frequency bands. An SD propagation may show up as propagating DC shifts across electrodes in the near-DC components or propagating depressions (power reductions) in higher frequency components (> 0.5 Hz). In ref. [24], Hartings et al. reported that EEG Delta band power, on average, depressed to 47% of its baseline during SD events observable in the EEG recordings, whereas other higher-frequency bands experienced less power reduction (i.e., Theta, Alpha, and Beta bands maintained around 60% or more of their baseline power). In addition, since around 81% of the total power of baseline EEG (without SD) was concentrated in the Delta band[24], the contrast between the background EEG power (baseline) and the maximum depressions during SD episodes was much higher for the Delta band in comparison to the higher-frequency bands. Therefore, we expected to observe a decreasing trend of WAVEFRONT performance as the function of frequency bands. To test this

hypothesis, we trained and validated WAVEFRONT based on the different frequency bands of near-DC, Theta, Alpha, and Beta by closely following the steps in the previous section. The average validation ROC curves for different frequency bands are shown in Fig. 6. Based on the results, in the same ROC region of PPV ≥ 0.50, the Delta band achieves the best detection performance (TPR = 0.74 ± 0.03, FPR = 0.015 ± 7.57 × 10⁻⁴), followed by Theta (TPR = 0.73 ± 0.031, FPR = 0.020 ± 0.0015), Alpha (TPR = 0.65 ± 0.039, FPR = 0.020 ± 0.0011), near-DC (TPR = 0.63 ± 0.013, FPR = 0.018 ± 7.05 × 10⁻⁴), and Beta (TPR = 0.59 ± 0.034, FPR = 0.016 ± 0.0011), all reported in 95% confidence intervals. This SD detection performance trend across different frequency bands is consistent with the expected outcome based on the reported results in[24] using visual inspection of SD events in the scalp EEG recordings. Although the propagating DC shifts are well-known signatures of SD waves in the ECoG recordings of the brain[8], WAVEFRONT has lower SD detection performance using near-DC components of scalp EEG signals (11% less TPR, 0.4% more FPR, and 9% less PPV) compared to the best performance among higher-frequency bands (i.e., the Delta band). Previous works also reported no propagating DC shifts[9,25], or

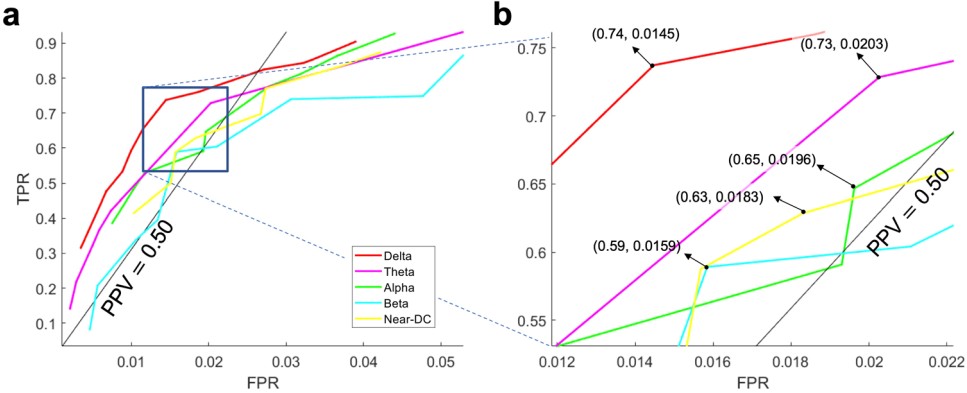

**Fig. 6 WAVEFRONT performance evaluation across different scalp EEG frequency bands. a** Receiver operating characteristic (ROC) curves of the average validation performance of WAVEFRONT in the detection of spreading depolarization (SD) events using noninvasive scalp electroencephalography (EEG) signals across different frequency ranges: [0.001, 0.01]Hz (near-DC, yellow curve), [0.5, 4]Hz (Delta, red curve), [4, 8]Hz (Theta, pink curve), [8, 12]Hz (Alpha, green curve), and [12, 30]Hz (Beta, cyan curve). The vertical axis indicates true positive rates (TPRs), and the horizontal axis shows false positive rates (FPRs). The black line indicates the positive predictive value of 0.50 (PPV), and the region above this line shows the operating points with PPV≥0.50, **b** Zoomed-in version of the ROC curves around the optimal validation operating points across different frequency bands. The best SD detection performance corresponds to the Delta band, followed by the Theta, Alpha, near-DC, and Beta bands.

very few consistent DC shift propagations in the scalp EEG signals, in comparison to the ECoG recordings of the SD events[24].

**Performance of WAVEFRONT in prediction of SD frequency.** In this section, we explore the performance of our method in predicting the frequency of SDs from the total minutes of detected SD events. This analysis is inspired by the recent work of Jewell et al.[10], where a linear regression was used to estimate the number of SDs in 24-hour time windows. In ref. [10], to automate ECoG-based SD detection, Jewell et al. developed a technique for real-time SD detection using ECoG signals by combining features from low-frequency bands (e.g., slow potential changes in 0.005–0.5 Hz) and high-frequency bands (e.g., reduction of envelope amplitude in 0.5–45 Hz). They reported a regression slope of ~ 0.79 with a 0.9% false alarm rate in 18 acute brain injury patients with DHC. In[10], the overall sensitivity was calculated by comparing the number of detected SDs to the number of ground truth SDs in 24-hour nonoverlapping time windows across patients. They used a linear regression for this comparison and reported a slope of ~ 0.79. The reported false positive rate (FPR) is a median value of the calculated FPR for each of the 24-hour time windows, where the negative events are defined as 20 min periods without any ground truth SDs.

The frequency of SD occurrence could serve as a metric to help clinicians make an informed decision about the choice of medications and/or invasive procedures for brain injury patients. Frequent occurrence of SDs in continuous recordings of the brain is correlated with worsening brain injury and poor outcomes in acute neurological conditions such as hemorrhage, ischemic stroke, and TBI[5,10,39,59]. Therefore, estimating the frequency and duration of SDs could be an important step toward personalized medicine.

WAVEFRONT's 74% detection rate of SD events is promising, but is it sufficient for noninvasive estimation of the frequency of SDs? To evaluate WAVEFRONT's performance for this purpose, we performed the following steps: (i) we extracted overlapping 30-hour time windows with a step size of 1-hour across all of the 12 patients. Time windows with poor EEG quality were ignored, i.e., windows with less than 20-hour of reliable (not masked out, see Methods) EEG signals across ≥5 ipsilateral scalp electrodes. There are $N_w = 153$ total time windows with good EEG quality (based on the definition provided above) in this dataset. (ii) For each time window, we applied WAVEFRONT to obtain the

temporal detection output $T_{out}$ (see Methods for details), where $T_{out} = 1$ indicates the detected SD events at the corresponding timepoint. (iii) We pruned and stitched together the detected intervals in each of the 30-hour windows. We removed isolated small detection intervals, which are less than 20 min long and separated from other detection events with more than a 4-hour temporal distance. After this pruning step, the remaining detected intervals were stitched together in a 4-hour sliding time window. This stitching process was at a very large temporal scale, in comparison to the original stitching process in the last step of WAVEFRONT (see Methods for details). We were not looking for the single-SD detection performance in this analysis, but the performance of WAVEFRONT in predicting SD frequency using the total duration of detections. Figure 7 shows the total duration of detected events for each of the 153 time windows (blue dots) as a function of the number of annotated SD events. The number of SD events in these time windows ranges from 0 to 75, and the total detection duration in each time window lies in the range of 0 to 26.85-hour. In this figure, the increasing trend of detection duration as a function of the frequency of SDs is as expected. In addition, some piecewise flat parts around intervals of 22 to 37 and 54 to 75 SDs are observed in Fig. 7, resulting from the under-detection of SD events in the time intervals. (iv) Finally, we measured the performance of WAVEFRONT in the estimation of the number of SDs from the total duration of detected SD events in the 30-hour time intervals. A square root regression model of $a\sqrt{x} + b$ was used, where the number of annotated SDs in each window is the independent variable, and the total duration of detections is the observation. We chose a square root model due to the sublinear increasing trend of the total duration of detections for the larger numbers of SDs (e.g., ≥25 SDs).

This sublinear increase rate of detection durations was caused by the under-detection of SDs, especially the highly clustered events, which underscores the limitation of WAVEFRONT in individual detection of SDs in such highly clustered events. We fitted the square root model to the observations using a least square regression.

Preliminary results, albeit with limited data, suggest that WAVEFRONT can reliably estimate the number of occurrences of SDs in long time intervals of 30-hour, with $R^2 \simeq 0.71$ and $RMSE = \sqrt{\frac{1}{N_w}\sum_{i=1}^{N_w}(x_i - \hat{x}_i)^2} \simeq 12.8$ SDs, using a square root regression, as shown in Fig. 7. $x_i$ is the number of annotated SDs

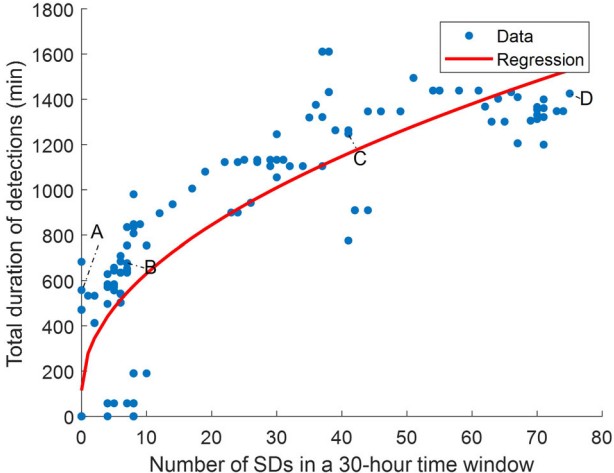

**Fig. 7 Performance of WAVEFRONT in prediction of SD frequency.** Each blue dot shows the total duration of detected spreading depolarization (SD) events using WAVEFRONT in 30-hour time windows after pruning small, isolated detection events and stitching together the remaining detection events. The expected increased trend of total detection duration as a function of the number of SD events was observed, with piecewise flat parts around 22–37, and 54–75 SDs, which are the clustering detection side effects in these time intervals. A square root regression model (red curve) was fitted and used to quantify the prediction performance ($R^2 \simeq 0.71$). Detected intervals in windows with a small (point A and B), medium (point C), and large (Point D) number of SDs are shown in Figs. 8 and 9.

in the i-th window, and $\hat{x}_i = \left(\frac{y_i - b}{a}\right)^2$ is the estimated number of SDs through the regression analysis based on the total duration of detections in each time window (i.e., $y_i$). Based on the results, WAVEFRONT can successfully discriminate between long windows of recordings with a large number of SDs (e.g.,> 40) and small number of SDs (e.g.,< 20), and estimate the frequency of SDs with a RMSE of less than 13 SDs. Such an analysis can also be used to assign the patient an SD score. Figures 8 and 9 show the detected intervals (marked with red strips on the bottom) along with the ground truth annotated SDs (dashed vertical lines) in some sample time windows with a small, medium, and large number of SDs. Figure 8a corresponds to point A in Fig. 7 and shows the detected false alarms in a patient without any SDs (patient 7), with a total detection duration of 558 min. The quality of the EEG recording in this time window is poor, with large portions of four of the electrodes' signals missing. This may explain the large number of false alarm detections. Figure 8b corresponds to point B and shows a 30-hour time window with seven SD events, where the WAVEFRONT algorithm successfully detects the isolated event, as well as the clustered scCSD events (see Methods for the definitions of SD annotations). Figure 9a and b show time windows with a larger number of SDs, in two patients with right (patient 3) and left (patient 12) DHCs. These windows correspond to points C and D in Fig. 7) with 41 and 75 SDs respectively, with highly clustered detection intervals. There are some missed SDs in between the two detection intervals in Fig. 9b, which may be due to the removed outlier artifact across all of the EEG electrodes.

*Why choose overlapping 30-hour windows?.* We intentionally chose a large window size of 30-hour to include windows with a large number of SDs (up to 75 SDs in a single window). This was important to explore the performance of WAVEFRONT for highly clustered SDs (see Figs. 7 and 9). The choice of 30-hour

window length was based on a heuristic approach to cover the widest possible range for the number of SDs across windows. The choice of overlapping windows in the SD frequency analysis was due to the limited available data and small number of patients in this study. This helped us have sufficient data points (blue dots in Fig. 7) across different numbers of SDs (the horizontal axis) for the regression analysis. However, the overlap introduces undesirable correlations in the data points for the regression analysis. This is a shortcoming due to the small dataset in this study. More accurate estimation of the number of SD events requires further improvements in the algorithm and a larger dataset with a wide variety of frequencies and types of SD events across patients.

## Discussion

In this study, we explored the feasibility and quantified the performance of automated noninvasive SD detection using continuous scalp EEG recordings from 12 severe TBI patients. These patients underwent DHCs and experienced 700 total SD propagation events over days ($95 \pm 42.2$-hour) of simultaneous EEG and ECoG recordings in ICUs. Intracranial signals were used for SD event temporal annotation. Our previously proposed WAVEFRONT algorithm[30], with appropriate modifications and improvements, achieves a reliable SD detection performance of 74% average cross-validation TPR (~ 13,000 of the ~ 17,000 total SD windows across the validation sets are detected), with less than 1.5% average cross-validation FPR (less than 7000 false alarms among the total 450,000 non-SD intervals) using Delta band scalp EEG signals. For the two patients without any annotated SD, the average false alarm rate is 1.7%, similar to the overall average of 1.5% cross-validation FPR. To understand the clinical implications of this, we evaluated the performance of WAVEFRONT in predicting of the number of SDs in long 30-hour time intervals using a square root regression. Preliminary results, albeit with limited data, suggest that WAVEFRONT achieves a promising performance (regression with $R^2 \simeq 0.71$) in the estimation of SD frequencies, despite a substantial number of false alarms. The SD detection performance in the Delta band was better than that in the Theta (73% TPR, 2.0% FPR), Alpha (65% TPR, 2.0% FPR), near-DC (63% TPR, 1.8% FPR), and Beta (59% TPR, 1.6% FPR) bands. This decreasing trend of SD detection performance is consistent with the existing understanding and literature, as the depth of SD depressions (i.e., percentage of maximum power reduction from the baseline power) reduces in higher-frequency bands, with the largest reported depression depth, and highest baseline level in the Delta band EEG[24]. However, although SPCs are the prominent signatures of SDs in DC or near-DC ECoG signals[8], WAVEFRONT's performance is worse when using near-DC EEG compared to the higher-frequency components in the Delta band. This may be due to the inherent limitation of EEG in capturing the slow DC shifts[24], or the limitation of WAVEFRONT in extracting SPCs. The estimated average propagation speed of the detected SD events in the EEG Delta band using WAVEFRONT is $3.35 \pm 0.05$ mm/min.

Real-time monitoring of patients with brain injuries is crucial to predict and prevent worsening brain injuries through SD detection at ICUs. Our modified WAVEFRONT algorithm in this paper can provide SD detection results for each 4-hour time window (epoch) with only 5 min computational delay (see Supplementary Note 2 for details). The WAVEFRONT algorithm achieves a reliable automated detection performance using scalp EEG. Preliminary evidence in SD frequency analysis suggests that WAVEFRONT achieves a promising performance in the prediction of SD frequency in long time intervals. Increasing evidence shows that SDs are reliable predictors of TBI patients' outcomes[5,10,19,60]. WAVEFRONT can potentially be used for

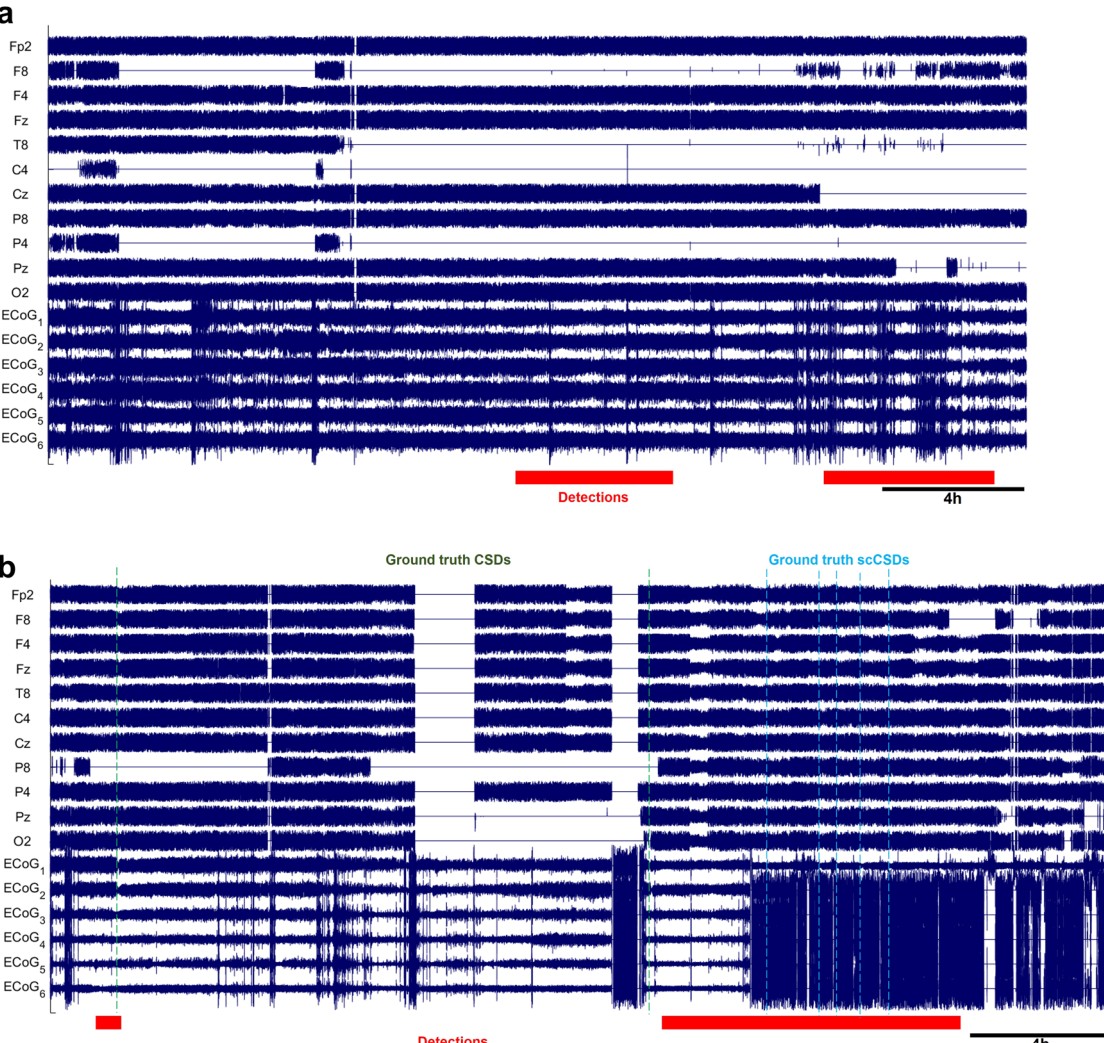

**Fig. 8 Detected SD intervals in time windows with small number of SDs.** Detection intervals (marked with red strips on the bottom) along with the ground truth annotated spreading depolarizations (SDs, dashed vertical lines) in long time windows, where the preprocessed ipsilateral electroencephalography (EEG) signals and electrocorticography (ECoG) signals are shown. EEG electrodes were ordered using the transverse montage (see Fig. 5a). Signals were normalized by their standard deviations for the illustrative purposes: **a** a 26.6-hour time window, corresponding to point A in Fig. 7, with no SD event in patient 7. However, there are large false alarm detection intervals with a total duration of 558 min, which may be explained by the poor quality of the EEG recording in this time window. **b** a 30-hour time window with seven SDs (two cortical spreading depressions (CSDs) and seven single-channel CSDs (scCSDs)), which corresponds to point B in the regression figure, and recorded from patient 6. WAVEFRONT successfully detects the the isolated CSD event as well as the clustered SD events toward the end of the window, with a total detection duration of 676min.

prognostication of worsening brain injury by providing a measure of SD frequency.

Validation of WAVEFRONT on DHC patients is a good starting point for noninvasive and automated SD detection because: a) intracranial ground truth for SDs can be obtained by placing ECoG electrodes during the DHC procedure, and b) head layers, including skull, meninges, cerebrospinal fluid (CSF), and scalp, have low-pass filtering or blurring effects on the scalp EEG signals. This makes the detection and tracking of narrow SD waves challenging[29,30]. This is less challenging in DHC patients due to the absence of the low-conductivity skull layer. Nevertheless, the challenge is considerable: i) relative to ECoG, the signal is more noisy and spatially low-pass filtered, and ii) as the SD wave propagates into the sulcus, its representation in the scalp EEG signals reduces substantially. This breaks the waves, as measurable by EEG, into disconnected components, which we call wavefronts[30]. Complex patterns of SDs (e.g., single gyrus[61,62], semi-planar[62–64], etc.) can make the noninvasive detection of

these waves even more challenging. WAVEFRONT addresses some of the difficulties in noninvasive detection of SD waves in EEG. It breaks down the challenging task of detecting the whole propagating SD wave in the brain using noisy and blurry filtered scalp EEG signals into simpler tasks of detection and classification of disjoint SD wavefronts. This overcomes the challenge related to the effects of sulci and gyri discussed above and enables the detection and tracking of complex patterns of propagation.

SD detection in patients with DHC is a clinically relevant problem. DHC is widely used for the management of severe TBI patients[35–37]. Around 60% of these patients experience SDs, mostly with a high frequency of occurrence[10,39], which increases the chance of worsening brain injury[5,10,19]. Every year in the United States, more than 1.2 million TBI patients experience worsening brain injury or death[23]. These patients form a large target population for continuous monitoring of SDs in ICUs. Scalp EEG in DHC patients provides wider spatial coverage of the brain than a locally placed intracranial strip of electrodes on the cortex. It also

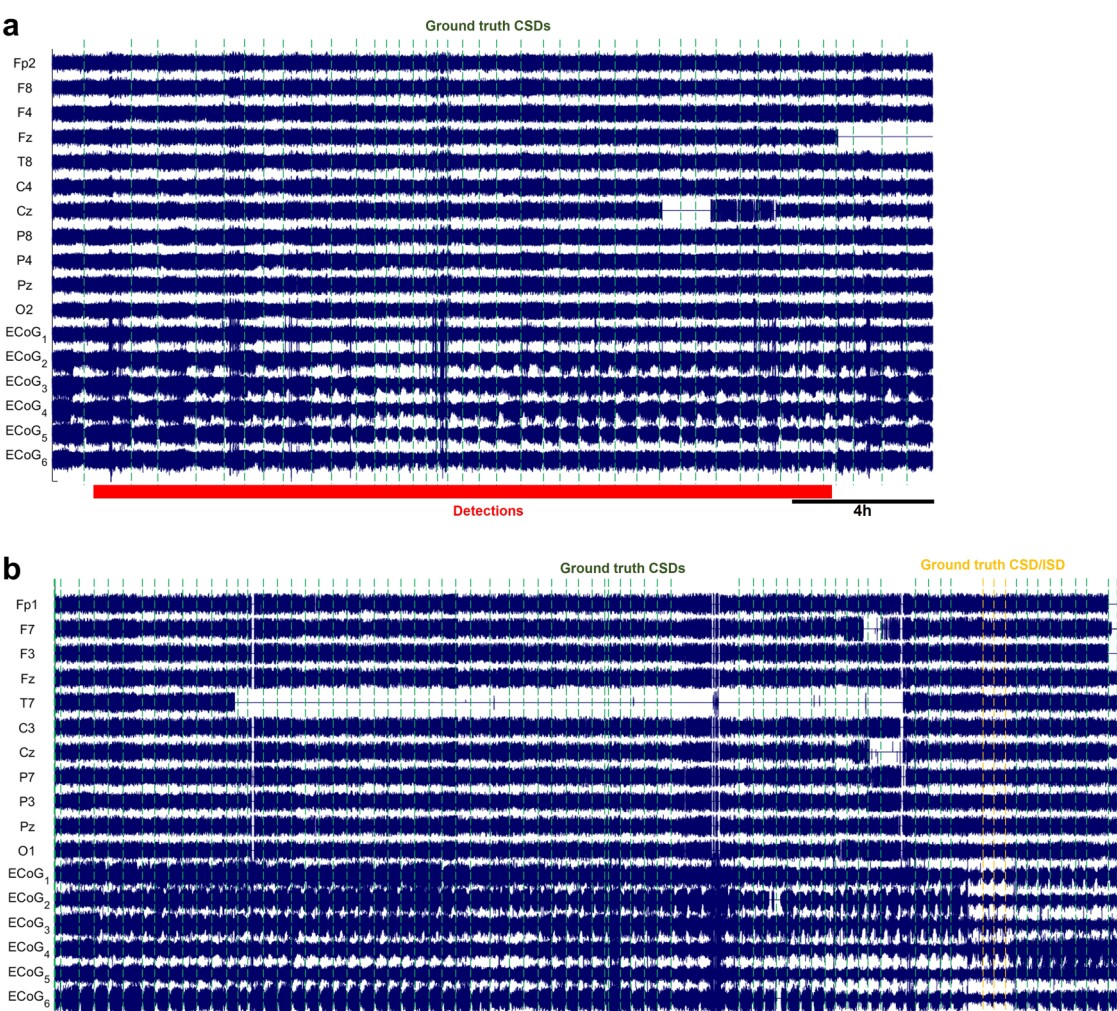

**Fig. 9 Detected SD intervals in time windows with large number of SDs.** Detection intervals (marked with red strips on the bottom) along with the ground truth annotated spreading depolarizations (SDs, dashed vertical lines) in long time windows, where the preprocessed ipsilateral electroencephalography (EEG) signals and electrocorticography (ECoG) signals are shown. EEG electrodes were ordered using the transverse montage (see Fig. 5a). Signals were normalized by their standard deviations for the illustrative purposes: **a** A 24.84-hour time window with 41 highly clustered SDs (point C in Fig. 7) in patient 3. **b** A 30-hour time window with the largest number of SD events, 75 (point D in Fig. 7), in patient 12. WAVEFRONT detects long intervals in (**a**) and (**b**), with the total duration of 1248 min and 1425 min.

has a higher spatial resolution in comparison to the EEG recordings from patients with an intact skull[35] and is less risky than intracranial electrode placement. Therefore, noninvasive detection and monitoring of SDs in severe TBI patients with DHC can help improve outcomes. In this paper, we only validated the performance of our technique using ipsilateral electrodes in the hemisphere with the removed skull, which limits the conclusions of this study to patients with DHC. A major motivation of our work for clinical management is to enable detection of SDs with an intact skull. The use of ECoG placed after DHC as a ground truth in this study was pragmatic, but limits the generalizability of our findings to intact skulls. Methods for detecting SDs using invasive electrodes with an intact skull have been established[9]. Electrodes can be placed through burr holes in the skull (e.g., stereo EEG electrodes) or over the cortex (e.g., subdural electrodes) with bone flap replacements (e.g., the dataset in ref. [9]).

There are limitations associated with the SD ground truth and dataset in this study: (i) due to the limited spatial coverage of the ECoG electrode strip, the temporal annotations may not reflect the actual temporal onset of each SD event, and some of the SDs may even be missed in the ground truth here because some of the

waves may have started to propagate from an origin far from the intracranial strip. However, ipsilateral EEG electrodes provide full spatial coverage of the DHC hemisphere. This ground truth limitation makes it infeasible to quantify the performance of WAVEFRONT in determining whether SDs are present or not in each window. This may explain the limitation of WAVEFRONT in discrimination between windows with and without SDs, as is shown in Fig. 7. Another contributing factor to this poor discrimination performance might be the inherent limitation of the WAVEFRONT algorithm. This requires further study with higher spatial coverage of subdural SD recordings. (ii) The depression width, temporal duration, and propagation speed of SD waves are unknown. This may result in slight over or underestimation of the actual performance of WAVEFRONT in the detection of SD events. Notably, the brains of the severe TBI patients in our dataset had structural abnormalities due to the injuries and/or surgical interventions for hematoma extraction, e.g. to alleviate swelling. These structural abnormalities can affect the pattern and speed of propagation of SDs. A ground truth with higher spatial and temporal coverage and accuracy is warranted to further explore these effects and assess the performance of

WAVEFRONT. (iii) Due to the small number of patients in this study, overfitting to the available SD events is inevitable (see Results for a detailed discussion on this issue). We expected WAVEFRONT to achieve a better average validation performance by using a larger dataset of TBI patients with multiple SD events across different varieties of propagation patterns (single-gyrus, semi-planar, ring-shape, etc.), different ranges of propagation speeds, and in different brain regions. Additionally, a larger dataset would enable us to provide stronger statistical guarantees for detection and discrimination. (iv) Low-density EEG of this dataset limits the performance of WAVEFRONT. Based on our reported simulation results in ref. [30], WAVE-FRONT can detect narrow SD wavefronts, even single-gyrus SD propagations, using a sufficiently high density of EEG electrodes on the scalp. Thus, higher-density EEG might be needed for milder TBIs with narrow SD wavefronts. Further studies are needed to explore the effects of scalp EEG electrode montages and density on noninvasive SD detection performance.

In addition to the limitations of the dataset and ground truth, WAVEFRONT has inherent limitations: (i) this algorithm suffers from under-detection of SD events because clustered SDs (more than two SDs in a time interval of 3-hour or less[10,19]) in EEG cannot be detected individually using our current approach. For clustered SDs, where the baseline power in the higher-frequency bands ($\geq 0.5$ Hz) is already suppressed, SPCs in the near-DC band (1–10 mHz) should be used for detecting and tracking of SDs. WAVEFRONT needs further improvements for this detection (ii) Although the average false alarm rate of $\sim 1.5\%$ may seem small, this number corresponds to the $\sim 33\%$ of the total detected events (i.e., out of the total detected events, around one-third of them are false alarms). This can be a serious limitation for diagnostic and monitoring goals, including cases where the risks and side effects of interventions and treatments are high, and a much lower false alarm rate is required. (iii) Finding the right set of stitching parameters in the SD frequency analysis is heuristic, and there is room for improvements. This is a future direction for this work. Again, increasing the size of the dataset can help reduce FPR further.

This work is an attempt to explore the feasibility and quantify the reliability of noninvasive SD detection in severe TBI patients using an automated algorithm. This can potentially be used for prognostication of worsening brain injury, paving the way toward personalized medicine.

## Data availability
Supplementary Data 1 contains source data for the main figures with numerical results in this paper. The dataset was obtained as part of a multicenter clinical study, ClinicalTrials.gov ID number NCT00803036. The anonymized raw EEG dataset may be made available upon request, contingent upon contractual obligations and data sharing and cooperative agreements.

## Code availability
WAVEFRONT was developed in MATLAB (R2018b), using standard toolboxes, and EEGLAB toolbox (v2019.0)[43]. All MATLAB code is made available online on GitHub[65] (https://doi.org/10.5281/zenodo.8210380).

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

## Acknowledgements

This work was supported, in part, by grants from the National Science Foundation (NSF), Chuck Noll Foundation for Brain Injury Research, the Office of the Assistant Secretary of Defense for Health Affairs through the Defense Medical Research and Development Program under Award No. W81XWH-16-2-0020, and the Center for Machine Learning and Health at CMU, under Pittsburgh Health Data Alliance. A Chamanzar was also supported by Neil and Jo Bushnell Fellowship in Engineering, Hsu Chang Memorial Fellowship, CMU Swartz Center for Entrepreneurship Innovation Commercialization Fellows program. Dr. Elmer's research time was supported by the National Institutes of Health (NIH) through grant 5K23NS097629. Opinions, interpretations, conclusions, and recommendations are those of the authors and are not necessarily endorsed by the Department of Defense. We thank Maysamreza Chamanzar, Shilpa George, Neil Mehta, David Okonkwo, and Praveen Venkatesh for helpful discussions.

## Author contributions

A.C., J.E., L.S., J.H., and P.G. initiated, designed, and executed the research. J.E., L.S., and J.H. supervised the data collection and provided the data. J.H. annotated the dataset. A.C. and P.G. developed the algorithms. A.C. developed the software tools necessary for analyzing the data. A.C., J.E., L.S., J.H., and P.G. wrote the manuscript.

## Competing interests

The authors declare the following competing interests: the authors filed a provisional patent on the technology, assigned to Carnegie Mellon University. P.G. is co-founder of Precision Neuroscopics Inc., a medical device company that intends to license the resulting patent from Carnegie Mellon University, and A.C. has equity in this company. All other authors declare no competing interests.
