## [Peer Review File · Communications Medicine]

Reviewers' comments:

Reviewer #1 (Remarks to the Author):

Title: Noninvasive, automated and reliable detection of spreading depolarizations in severe traumatic brain injury using scalp EEG

The authors proposed a WAVEFRONT technique for the automated and reliable detection of spreading depolarization (SD) in severe traumatic brain injury (TBI) patients using scalp EEG. For this purpose, low-density scalp EEG (19 electrodes) from 12 patients with severe TBI who have undergone for decompressive hemicraniectomy were considered. Further, the intracranial EEG was also used as a ground truth for the detection of SD events. The research addresses some of WAVEFRONT shortcomings such as fixed set of parameters, implicit assumption on power levels and sensitivity to amplitude outliers by designing a training and validation framework. The proposed approach detects the SD with a maximum TPR of 74% in the Delta band and the TPR found to decrease in higher frequencies such as theta, alpha, and beta bands.

It is an interesting paper that presents a very thorough analysis of the results. There seems to be a significant contribution from them. There are, however, a few major comments to be addressed before the final acceptance.

Comments:

1. What are the scalp EEG characteristics of SD and how is it different from intracranial EEG? What patterns does the proposed WAVEFRONT detect?
2. The results show that the low frequency (delta band) is more informative than others. It would be great if you could mention the complete details of band pass filtering characteristics (Filter type, order, pass and stop band characteristics)
3. In reference 10, the detection rate was found to be around 79% in 24-hour non-overlapping time windows across patients. So it is worth mentioning the influence of window length, shape, and overlap in the context of WAVEFRONT. Does WAVEFRONT alter the detection performance, if yes what would be the optimum choice, and shape?
4. Is it possible to detect the complex patterns of SDs using WAVEFRONT?
5. It is reported that scalp EEG from 19 electrodes (10-20 electrode system) are considered for this research. Whether different montage connections influence the detection performance? It is important to consider montages when we worry about spatial resolution.
6. Will spatial subsampling affect the detection performance of WAVEFRONT?

Reviewer #2 (Remarks to the Author):

This manuscript addresses a highly significant issue of non-invasive detection of spreading depolarization (SD) from patients with brain injury. Work in the last two decades points to SD being a principal driver of injury progression and intervention trials targeting SDs are beginning. A large gap in the field is reliable non-invasive detection of SD in patients. At present, SDs can only be detected in the small number of patients undergoing surgical procedures (usually decompressive craniectomy) that also allow placement of subdural strip electrodes to monitor SD. The present work seeks to improve upon methods to detect SD with conventional EEG electrode arrays, placed over the scalp. The authors have previously published an algorithm (WAVEFRONT) for automated non-invasive detection of SD based on EEG simulations (Chamanzar et al., 2019). The present work builds upon that valuable report by adding refinements to the WAVEFRONT algorithm and testing the approach on EEG data collected from a set of 12 severe TBI patients. Comparison with gold-standard ECoG strip recordings provided good performance of the algorithm, particularly in the delta band. This is a very good accomplishment, and provides a good foundation for future testing with larger data sets. However as noted below, there are issues related to clinical applicability that would be helpful to address in more detail in the text and analysis, particularly as there do appear to be some real

barriers to non-invasive EEG detection through intact skull that likely remain, and these somewhat limit the potential impact of this report on the field. There are also some general concerns related to the presentation of the manuscript that could be help readers.

1. Overall the manuscript is very long (>12,000 words), more than double the recommended ~5000 words for Articles in this journal. The Introduction section is very long (~2400 words) and delves into detailed discussion of the technical approach, limitations of the work, and comparison with other methods and algorithms. There is also significant repetition of Introduction text in the following sections. The Introduction should be substantially shortened and restructured, to follow the Journal's formatting guide of providing a background and rationale for the work, and then finishing with a brief summary of the major results and conclusions. Some repetitive sections within the Results (that are more appropriate for Introduction) should also be removed.

2. Related to the point above, the Authors could consider moving sections of text descriptions of methods into a Supplemental section. One example would be detailed description of methods previously published in reference [28] (pages 8,9 & 11), and other detailed methods.

3. A major goal for clinical management is to be able to monitor SD in patients without craniectomy – ie those patients who don't already have the access available for subdural strip recording. A significant limitation of the current study is that the detection algorithm was only used on EEG data acquired from scalp recordings over the craniectomy. The authors acknowledge this limitation and appropriately explain that recording from ipsilateral cortex provides the best test against ground truth ECoG recordings from electrode strips that had been placed over the exposed cortex. However, to address the key question of whether the method works through intact skull, it would be ideal to have ground truth recordings from electrodes placed into (or onto) the cortex through the intact skull. Although these recordings are not available in the clinical data set used here, such methods have been in development and have been published. It would be helpful to discuss this point more explicitly, as a required next validation step to achieve the main goal highlighted in the Introduction.

4. Related to the point above, the degree to which EEG signals are enhanced by removal of the skull is an important issue for potential future clinical applicability. However the influence of the skull on the amplitude of signals is only rather vaguely reported. Figure 2 shows an example of contralateral vs ipsilateral signals, and makes the general case that there is some attenuation based on visual inspection of the traces shown. It would be helpful to have group data from more patients to assess the extent of this attenuation across subjects, and in relation to the placement of electrodes over the craniectomy (or not). It is confusing that on page 8, that the authors indicate that for Figure 2 (on the ipsilateral side) "electrodes which are far from the site of surgery have lower baseline power, as compared with electrodes right on top of regions with a missing skull". This description of electrode placement with relation to the craniectomy is difficult to appreciate with the 2 dimensional rendering shown in Fig 2a, and it would also be helpful to quantify the degree to which this signal diminution occurs in the group data, with distance from the craniectomy boundary on the ipsilateral side.

5. It was pointed out in their previous publication [28] that the WAVEFORM algorithm is not real-time and that future work would extend this algorithm to enable real-time detection – something that was noted as an important factor for clinical applicability. Can the authors comment on processing time required to complete a detection map of SD with the modified WAVEFORM processing described here, and also indicate whether and how significant acceleration towards real-time detection could be expected in the future?

6. Page 14 and Fig 9 show a distribution of estimated speed of propagation of SD events calculated from the WAVEFORM algorithm. It would be useful to compare this with ECoG recordings, but it is noted (p9) that "no ground truth is available for the speed of propagation of annotated SD events in the dataset". Couldn't this be calculated from the propagation of events across electrodes of the ECoG strip? This question seems interesting in light of the comment (p13) that "DHC patients have highly

concave scalp surface on the hemisphere with missing skull". It would be helpful to add a comment on whether or not the detected rates and/or patterns of propagation would be as easy to track with WAVEFORM in brain that better retains the normal structure of sulci and gyri.

7. Per Journal style recommendations, remove use of italics of emphasis.

8. Page 4: Figure 1, the description of red and green arrows should be added to the figure legend, and related more clearly to the text describing the location of DCH and evacuation of hematoma.

9. Bottom of page 4/top of page 5: It is stated that a unique SD wave was "annotated by consideration of, and manifested across all electrodes of the ECoG strip". This should be modified to be consistent with the rest of the paragraph, which describes events classified as SDs that are not seen in all electrodes (or even in a single electrode).

Reviewer #3 (Remarks to the Author):

Review for Communications Medicine

Title: Noninvasive, automated and reliable detection of spreading depolarizations in severe traumatic brain injury using scalp EEG

Authors: Chamanzar A et al.

This is an interesting paper that sets out to determine whether there is possible a noninvasive and reliable detection of spreading depolarizations in severe brain injury using a scalp EEG. The authors modified the previously developed WAVEFRONT algorithm and used it to detect spreading depolarizations in 12 patients with severe traumatic brain injury who underwent decompressive hemicraniectomy. The hypothesis was that the automated detection of the SDs using noninvasive electroencephalography (EEG) would be possible. At the same time a SD-detection using ECoG strip electrodes in the decompressive hemicraniectomy side was performed. The quantification of performance was based in the accuracy detection of spreading depolarizations, including true positive rate (TPR), false positive rate (FPR) and the estimation of the frequency of SDs. The authors conclude that WAVEFRONT can achieve a very good performance in the estimation of the SD detection. The paper is generally well written and the authors are from a well-known team in the field of neurosurgery and critical care medicine. However, there are some concerns regarding the methodology and presentation of the study. The conclusions are original and convincing. The impact hast to be considered as moderate.

Major points:

-As mentioned by the authors the small sample size. With only 12 patients is difficult to have a safe conclusion.

- The fact the authors used only ipsilateral scalp EEG electrodes. May that influence the conclusions?

-The performance of the WAVEFRONT to detect cluster of SDs is not clear.

Noninvasive, automated and reliable detection of spreading depolarizations in severe traumatic brain injury using scalp EEG

Alireza Chamanzar, Jonathan Elmer, Lori Shutter, Jed Hartings, Pulkit Grover

Response letter cover:

We thank the reviewers and the editor of our paper for the positive feedback and constructive comments they provided through the peer review process. We are pleased that they have found the paper “interesting” and one “that presents a very thorough analysis”. In this revised version of the manuscript, we attempted to fully address the reviewers’ comments. We have also performed additional analyses based on their suggestions and to provide additional evidence for our claims. We are grateful for these suggestions, which have, in our view, resulted in substantial improvement in our paper. Following is an outline of the major changes in this revision and, below which, we provide detailed responses to each of the reviewers’ comments:

- **Exploring the potential of real-time SD detection using WAVEFRONT:** Following the Reviewer#2’s comment 5, we measured the computational delays at various steps of our modified WAVEFRONT algorithm, and provided a detailed discussion on the potential of WAVEFRONT for real-time monitoring of SDs. WAVEFRONT can detect SDs for each 4-hour time window (epoch) with only ~5min computational delay. These details are included in Supplementary Note E.
- **Cutting down and restructuring the paper:** We significantly cut down the length of the paper and restructured the sections and figures to meet the format requirements. Some sections and figures are moved to the Supplementary Materials. Please see our response to Reviewer#2’s comments 1 and 2.
- **Detailed justification and discussion provided on the choice of window size and shape for the SD frequency analysis:** More details on this are provided in our response to Reviewer#1’s comment 3.
- **New statistical analysis for hemispheric comparison of average baseline power on the scalp:** For the 12 patients, we estimated the average baseline power of scalp EEG signals for the ipsilateral and contralateral hemispheres, and assessed the statistical significance of this hemispheric difference across different time windows. Based on the results, for all of the 12 patients in the dataset, there is a significant difference ($p < 1e-8$) between the ipsilateral and contralateral average power ($\bar{P}_{contra} < \bar{P}_{ipsi}$). Please see our response to Reviewer#2’s comment 4.

In addition, some minor modifications have been made in the revised manuscript to clarify the claims, and to add further discussion on the limitations and the future directions of the paper. We are grateful for the reviewers’ feedback, which made the content of the paper easier to understand and follow.

Following are our point-by-point responses to the reviewers’ comments:

Color codes:

- Reviewers' comments/questions
- Authors' responses
- **Bold fonts: Modified/added texts in the revised manuscript**

Reviewers' comments:

Reviewer #1 (Remarks to the Author):

The authors proposed a WAVEFRONT technique for the automated and reliable detection of spreading depolarization (SD) in severe traumatic brain injury (TBI) patients using scalp EEG. For this purpose, low-density scalp EEG (19 electrodes) from 12 patients with severe TBI who have undergone for decompressive hemicraniectomy were considered. Further, the intracranial EEG was also used as a ground truth for the detection of SD events. The research addresses some of WAVEFRONT shortcomings such as fixed set of parameters, implicit assumption on power levels and sensitivity to amplitude outliers by designing a training and validation framework. The proposed approach detects the SD with a maximum TPR of 74% in the Delta band and the TPR found to decrease in higher frequencies such as theta, alpha, and beta bands.

It is an interesting paper that presents a very thorough analysis of the results. There seems to be a significant contribution from them. There are, however, a few major comments to be addressed before the final acceptance.

We thank the reviewer for the positive assessment of the manuscript and our contribution in the field.

Comments:

1. What are the scalp EEG characteristics of SD and how is it different from intracranial EEG?

Response to 1-Part1) This is a very important question. Scalp EEG characteristics of SD include: (i) In scalp EEG, SDs appear as propagating power depressions (reductions) across electrodes on the scalp, which are dominant in the Delta band, while in intracranial EEG, SDs show up as a slow near-DC negative shifts (in 1-10 mHz), i.e., slow potential changes (SPCs), often accompanied by power reductions in higher frequency bands (>0.5Hz), which are called cortical spreading depressions (CSDs). In highly clustered SDs, high-frequency activity is already suppressed, and only SPCs happen at ECoG electrodes, which are called isoelectric spreading depolarizations (ISDs); (ii) SDs as observed on scalp EEGs have a slightly slower propagation speed (e.g. from our inference, the speed on the scalp is 0.9-6.8mm/min); (iii) The power depressions at each electrode have slower falling and rising edges in comparison to the

sharp edges of power reductions/depressions in intracranial EEG (see Fig. 9d for a side-by-side comparison of SD power depressions in EEG and ECoG).

What patterns does the proposed WAVEFRONT detect?

Response to 1-Part2) WAVEFRONT detects and tracks propagating power depressions (reductions) in all high-frequency bands ($>0.5\text{Hz}$), achieving its best performance in the Delta band (74% TPR and 1.5% FPR). However, our WAVEFRONT algorithm as-is doesn't perform as well in extraction and detection of SPCs on scalp (70% TPR, $>2.1\%$ FPR, see Fig. 7 for more details on frequency band analysis) and suffers from underdetection of highly clustered SDs (i.e., SDs with only near-DC shift manifestations in 1-10mHz). Modifying and improving WAVEFRONT for detection and tracking of near-DC components of SDs is a future direction for this work.

We have now added a short discussion on this in Section IV:

“...(i) this algorithm suffers from underdetection of SD events since clustered SDs (more than two SDs in a time interval of 3 hours or less [10, 19]) in EEG cannot be detected individually using our current approach. **For clustered SDs, where the baseline power in the higher frequency bands ($\geq 0.5\text{ Hz}$) is already suppressed, SPCs in the near-DC band (1–10 mHz) should be used for detection and tracking of SDs. WAVEFRONT needs further improvements for this detection,...**”

2. The results show that the low frequency (delta band) is more informative than others. It would be great if you could mention the complete details of band pass filtering characteristics (Filter type, order, pass and stop band characteristics)

Response to 2) In this study, we used a zero-phase Hamming windowed sinc FIR filter. The filter function is implemented in the EEGLAB toolbox [2*] in MATLAB. We used a filter order of 10,000 to have a sharp frequency cutoff, which helps us retain the near-DC components (0.01,0.1Hz). The transition bandwidth (i.e., the bandwidth between the bandpass and stopband cutoff frequencies) is around 0.02Hz. To address this comment, we included these details in the Supplementary Note B, EEG preprocessing pipeline:

“...To be able to evaluate the performance of our SD detection algorithm in different frequency bands, we bandpass filter the EEG signals in different frequency ranges, namely, [0.01,0.1] Hz (near-DC), [0.5,4] Hz (Delta), [4,8] Hz (Theta), [8,12] Hz (Alpha), and [12,30] Hz (Beta) using a Hamming-windowed sinc finite impulse response (FIR) filter. The filter order is 1000 with a transition bandwidth of $\sim 0.02\text{ Hz}$.”

3. In reference 10, the detection rate was found to be around 79% in 24-hour non-overlapping time windows across patients. So it is worth mentioning the influence of window length, shape,

and overlap in the context of WAVEFRONT. Does WAVEFRONT alter the detection performance, if yes what would be the optimum choice, and shape?

Response to 3) Thank you so much for this important question. As a clarification, in reference 10 the reported performance metric is the regression slope of 0.79, which we reported as the detection rate by mistake. This issue is resolved in the revised paper. We answer this question in two parts: (i) The effect of window length on WAVEFRONT's performance: for window size of less than 4 hours, the pruning and stitching step of WAVEFRONT becomes less effective in pruning the false positive isolated events and stitching together the nearby detection events. This results in overdetection of SD events (higher false positive rate) and SD frequencies in a given time window. For larger time windows, the algorithm is not expected to be affected by the length of the window; (ii) The effect of window overlap on WAVEFRONT's performance: the overlaps between windows introduce correlations in the data points, which is not desirable for the regression analysis and results in overfitting: when independent variables (i.e., number of SDs in different windows) are correlated, the estimated coefficients in the regression model become very sensitive to small changes in the data which makes the regression model less generalizable to new observations. This is a data based multicollinearity [3*]. Therefore, overlap adversely affects the SD frequency prediction of WAVEFRONT.

We intentionally chose a large window size of 30 hours to include windows with a large number of SDs, with up to 75 SDs in a single window. This helps us assess the performance of WAVEFRONT for highly clustered SDs, as discussed in Section IIID and IV and illustrated in Fig. 8 and 9. We chose overlapping windows for our SD frequency (regression) analysis (see Section IIID) because of the limited data and small number of patients in this study. The overlap helps us have sufficient data points (blue dots in Fig. 8) across different numbers of SDs (the horizontal axis), and obtain a smooth curve, for the regression analysis. However, the overlaps between 30-hour windows introduce correlations in the data points, which is not desirable for the regression analysis and results in overfitting: when independent variables (i.e., number of SDs in different windows) are correlated, the estimated coefficients in the regression model become very sensitive to small changes in the data which makes the regression model less generalizable to new observations. This is a data based multicollinearity [3*]. This shortcoming of our analysis is acknowledged in the paper in Sections III and IV, and is precisely the reason why the analysis is described as "preliminary". For a large cohort of patients with a wide variety of frequency of SDs, non-overlapping windows can be used. We have mentioned the limitation of the dataset in this study in Section III:

"Preliminary results, albeit with limited data, suggest that WAVEFRONT can reliably estimate the number of occurrences of SDs in long time intervals of 30-hour,...."

In addition, to better address this comment, we added the following paragraph at the end of Section III to discuss the choice of window length and overlap in the regression analysis:

"Why choose overlapping 30-hour windows? We intentionally chose a large window size of 30-hour to include windows with a large number of SDs, up to 75 SDs in a single window. This is important to explore the performance of WAVEFRONT for highly clustered SDs (see Fig. 8 and 9). The choice of 30-hour window length is based on a heuristic approach to cover the widest possible range for

the number of SDs across windows. The choice of overlapping windows in the SD frequency analysis is due to the limited available data and small number of patients in this study. This helps us to have sufficient data points (blue dots in Fig. 8) across different numbers of SDs (the horizontal axis) for the regression analysis. However, the overlap introduces undesirable correlations in the data points for the regression analysis. This is a shortcoming due to the small dataset in this study. More accurate estimation of the number of SD events requires further improvements in the algorithm and a larger dataset with a wide variety of frequencies and types of SD events across patients.”

4. Is it possible to detect the complex patterns of SDs using WAVEFRONT?

Response to 4) This is a very important question. Based on the results in this study, the features that mostly drive the noninvasive detection of SDs on scalp include a Delta band power depression (power reduction in [0.5,4]Hz) that propagates across electrodes at a speed of 0.9 to 6.8 mm/min (see our response to comment 1 of Reviewer#1). The algorithm doesn’t make any other assumptions about the pattern of SD, especially on the shape of the wavefronts. Therefore, we expect WAVEFRONT to perform as well for complex patterns of SDs.

Based on our simulation results in [4*], WAVEFRONT detects and tracks complex patterns of SDs, including semi-planar wavefronts and single-gyrus propagations of SDs. This is because WAVEFRONT detects SD *wavefronts* individually, and stitches them together over time and space to more reliably detect (and possibly track) SDs. Notably, those results assumed a sufficiently high density of EEG electrodes on the scalp for detection of SD wavefronts as narrow as 5mm wide. However, in analyzing the real dataset that we used in this study there are only a limited number of patients with mostly wider, longer, and more frequent SDs due to the severity of their head injuries. Also, a single strip of ECoG electrodes does not allow us to assess the whole SD pattern in the brain. Therefore, further studies are needed to explore the performance of WAVEFRONT on patients with milder injuries, with a wide variety and more complex patterns of SDs, e.g., narrow single-gyrus propagations, semi-planar SDs, etc. We have mentioned these points in Section IV, Discussions and Conclusions:

“...We expect WAVEFRONT to achieve a better average validation performance by using a larger dataset of TBI patients with multiple SD events across different varieties of propagation patterns (single-gyrus, semi-planar, ring-shape, etc.), different ranges of propagation speeds, and in different brain regions. In addition, a larger dataset would enable us to provide stronger statistical guarantees for detection and discrimination, and (iv) low-density EEG, with a small number of electrodes was used in this dataset, limits the performance of WAVEFRONT. Based on our reported simulation results in [28], WAVEFRONT can detect narrow SD wavefronts, even single-gyrus SD propagations, using a sufficiently high density of EEG electrodes on the scalp. Thus, higher density EEG might be needed for milder TBIs with narrow SD wavefronts...”

5. It is reported that scalp EEG from 19 electrodes (10-20 electrode system) are considered for this research. Whether different montage connections influence the detection performance? It is important to consider montages when we worry about spatial resolution.

Response to 5) The EEG cap electrode density and montages indeed play an important role in detection and tracking of SD waves in the brain. E.g., in our earlier work [4*], we explored the performance of noninvasive SD detection using different densities of EEG on the scalp. Based on simulation results in [4*], WAVEFRONT using sufficiently high density of EEG (340 electrodes) can detect and track SDs, even the narrowest width of depression of 5mm (power reduction in the higher frequency bands, i.e., >0.5Hz), while the detection performance of a low-density EEG cap with 40 electrodes is limited to SDs with width greater than 1.1cm. In general, higher densities of EEG provide higher spatial resolution [5*,6*]. Further experiments and studies are needed to further explore the influence of EEG montage and density on noninvasive SD detection performance. To address this comment, we added the following in Section IV:

“...and (iv) low-density EEG with a small number of electrodes was used in this dataset, which limits the performance of WAVEFRONT. Based on our reported simulation results in [2], WAVEFRONT can detect narrow SD wavefronts, even single-gyrus SD propagations, using a sufficiently high density of EEG electrodes on the scalp. Thus, higher density EEG might be needed for milder TBIs with narrow SD wavefronts. Further studies are needed to explore the effects of scalp EEG electrode montages and density on the noninvasive SD detection performance.”

6. Will spatial subsampling affect the detection performance of WAVEFRONT?

Response to 6) Spatial subsampling is definitely an important factor in the detection performance of SDs and also the computational cost of using WAVEFRONT. For a very low spatial sampling rate on the extracted binary images on scalp (I_BW, see Fig. 4), wavefronts of narrow SDs might be missed. The reason is that, through the subsampling process, pixels of a very small connected component (white regions) might not be sampled in the resulting image with lower spatial resolution. If the SD is wide enough, this issue can be avoided by carefully choosing the parameters of the interpolation kernel (standard deviation, σ) and the spatial subsampling rate so that the corresponding pixels of each electrode location on the scalp have a representation in the lower resolution binary image. In this study, due to the very low density EEG cap with an average inter-electrode distance of ~5.4cm, we used a 2D Gaussian kernel with a large standard deviation of 2.6cm to obtain a smooth transition of SD wavefronts across electrodes, even in a subsampled image, without missing any connected component/wavefront through the subsampling process. To better address this comment, we added the following sentence in Supplementary Note C:

“...We carefully chose the parameters of the interpolation kernel (standard deviation σ) and the spatial subsampling rate so that the corresponding pixels of each electrode location on the scalp have a representation in the lower resolution binary image.

Therefore, we do not expect to have missed any connected component (wavefront) in the lower resolution binary image.”

References for our answers to the comments of Reviewer#1

[1*] Hartings, J.A., Wilson, J.A., Hinzman, J.M., Pollandt, S., Dreier, J.P., DiNapoli, V., Ficker, D.M., Shutter, L.A. and Andaluz, N., 2014. Spreading depression in continuous electroencephalography of brain trauma. *Annals of neurology*, 76(5), pp.681-694.

[2*] A. Delorme and S. Makeig. EEGLAB: an open source toolbox for analysis of single-trial EEG dynamics including independent component analysis. *Journal of neuroscience methods*, 134(1):9–21, 2004.

[3*] Goldstein, R., 1993. Conditioning diagnostics: Collinearity and weak data in regression.

[4*] A. Chamanzar et al. An algorithm for automated, noninvasive detection of cortical spreading depolarizations based on EEG simulations. *IEEE. Trans. Biomed. Eng.*, 66(4):1115–1126, 2018.

[5*] P. Grover and P. Venkatesh. An information-theoretic view of EEG sensing. *Proceedings of the IEEE*, 105(2), pp.367-384, 2016.

[6*] A. K. Robinson, et al. Very high density EEG elucidates spatiotemporal aspects of early visual processing. *Scientific reports*, 7(1), 16248, 2017.

Reviewer #2 (Remarks to the Author):

This manuscript addresses a highly significant issue of non-invasive detection of spreading depolarization (SD) from patients with brain injury. Work in the last two decades points to SD being a principal driver of injury progression and intervention trials targeting SDs are beginning. A large gap in the field is reliable non-invasive detection of SD in patients. At present, SDs can only be detected in the small number of patients undergoing surgical procedures (usually decompressive craniectomy) that also allow placement of subdural strip electrodes to monitor SD. The present work seeks to improve upon methods to detect SD with conventional EEG electrode arrays, placed over the scalp. The authors have previously published an algorithm (WAVEFRONT) for automated non-invasive detection of SD based on EEG simulations (Chamanzar et al., 2019). The present work builds upon that valuable report by adding refinements to the WAVEFRONT algorithm and testing the approach on EEG data collected from a set of 12 severe TBI patients. Comparison with gold-standard ECoG strip recordings provided good performance of the algorithm, particularly in the delta band. This is a very good accomplishment, and provides a good foundation for future testing with larger data sets. However as noted below, there are issues related to clinical applicability that would be helpful to address in more detail in the text and analysis, particularly as there do appear to be some real barriers to non-invasive EEG detection through intact skull that likely remain, and these somewhat limit the potential impact of this report on the field. There are also some general concerns related to the presentation of the manuscript that could be help readers.

Thank you for the positive evaluation of our manuscript and your very helpful comments on improving the text and analysis.

1. Overall the manuscript is very long (>12,000 words), more than double the recommended ~5000 words for Articles in this journal. The Introduction section is very long (~2400 words) and delves into detailed discussion of the technical approach, limitations of the work, and comparison with other methods and algorithms. There is also significant repetition of Introduction text in the following sections. The Introduction should be substantially shortened and restructured, to follow the Journal's formatting guide of providing a background and rationale for the work, and then finishing with a brief summary of the major results and conclusions. Some repetitive sections within the Results (that are more appropriate for Introduction) should also be removed.

Response to 1) Thanks for this comment on the paper length, format, and organization of the sections and the useful suggestions on how to cut it down. In this revision, to address these comments and fit within the recommended limits, we pushed most of the detailed description of our methods to the Supplementary Note B and C, restructured and shortened the Introduction, so that it starts with a background and rationale for the work, and then finishes with a brief summary of the major results and conclusions. In addition, we reduced repetition in the Results section as well as elsewhere in the manuscript, and moved Fig. 3, 5, 6, 7, and 11 to the Supplementary Materials to meet the limit on the maximum possible number of figures of 10.

2. Related to the point above, the Authors could consider moving sections of text descriptions of methods into a Supplemental section. One example would be detailed description of methods previously published in reference [28] (pages 8,9 & 11), and other detailed methods.

Response to 2) Thanks for this great suggestion. Based on this comment, we pushed most of the detailed description of our methods to the Supplementary Note C.

3. A major goal for clinical management is to be able to monitor SD in patients without craniectomy – ie those patients who don't already have the access available for subdural strip recording. A significant limitation of the current study is that the detection algorithm was only used on EEG data acquired from scalp recordings over the craniectomy. The authors acknowledge this limitation and appropriately explain that recording from ipsilateral cortex provides the best test against ground truth ECoG recordings from electrode strips that had been placed over the exposed cortex. However, to address the key question of whether the method works through intact skull, it would be ideal to have ground truth recordings from electrodes placed into (or onto) the cortex through the intact skull. Although these recordings are not available in the clinical data set used here, such methods have been in development and have been published. It would be helpful to discuss this point more explicitly, as a required next validation step to achieve the main goal highlighted in the Introduction.

Response to 3) We thank the reviewer for this important and critical comment. Indeed, a crucial next step for this study is to explore the feasibility of noninvasive and automated SD detection in patients with intact skulls. Skull is a very low-conductive layer in the head (the brain-to-skull conductivity ratio is reported to be around 80 [1*,2*]) and it attenuates the brain signals, with worse attenuation of the activity in high spatial frequencies, as they reach to the scalp through

the head layers. This will make the detection and tracking of SD waves challenging, and even more so for narrow SD waves [3*,4*]. Indeed, as the reviewer notes, this is less challenging in DHC patients due to the absence of the very low conductivity skull layer.

Methods for intracranial monitoring of SDs with an intact skull have been published [5*]. We have now cited a recent work [5*], where the authors monitored 15 TBI and 20 aneurysmal SAH (aSAH) patients, with simultaneous invasive and noninvasive recordings, some of whom received craniotomy. Availability of such datasets is exciting to us as this is the natural next step. To address this comment in the paper, we added the following sentence in the Discussions and Conclusions:

“... A major motivation of our work for clinical management is to enable detection of SDs with an intact skull. The use of ECoG placed after DHC as a ground truth in this study was pragmatic, but limits the generalizability of our findings to intact skulls. Methods for detecting SDs using invasive electrodes with an intact skull have been established [9]. Electrodes can be placed through burr holes in the skull (e.g., stereo EEG electrodes) or over the cortex (e.g., subdural electrodes) with bone flap replacements (e.g., the dataset in [9]). ”

4. Related to the point above, the degree to which EEG signals are enhanced by removal of the skull is an important issue for potential future clinical applicability. However the influence of the skull on the amplitude of signals is only rather vaguely reported. Figure 2 shows an example of contralateral vs ipsilateral signals, and makes the general case that there is some attenuation based on visual inspection of the traces shown. It would be helpful to have group data from more patients to assess the extent of this attenuation across subjects, and in relation to the placement of electrodes over the craniectomy (or not). It is confusing that on page 8, that the authors indicate that for Figure 2 (on the ipsilateral side) “electrodes which are far from the site of surgery have lower baseline power, as compared with electrodes right on top of regions with a missing skull”. This description of electrode placement with relation to the craniectomy is difficult to appreciate with the 2 dimensional rendering shown in Fig 2a, and it would also be helpful to quantify the degree to which this signal diminution occurs in the group data, with distance from the craniectomy boundary on the ipsilateral side.

Response to 4) We thank the reviewer for the suggestion of quantifying the signal baseline power reduction as a function of the distance from the boundary of the craniectomy site. Such an analysis can provide crucial information for future studies on noninvasive SD detection. Unfortunately, in this study we don't have access to the exact location of the craniectomy boundaries in these TBI patients: rendering a 3D model of the skull and head requires multiple slices of structural MRI scans, which weren't available for any of these patients (see our recent work in [8*] for examples of 3D head models extracted from high-resolution MRI scans).

However, to address this comment and provide a crude level quantification for this EEG baseline power enhancement in patients with craniectomy, we perform a hemispheric comparison of average baseline power for the 12 patients: (i) we choose non-overlapping 4-hour windows of scalp EEG recordings. The length of windows are arbitrarily chosen, but long

enough for average power estimation and short enough not to get affected by the baseline power variation over time, (ii) for each window, the average baseline power is estimated at each scalp electrode, (iii) the estimated baseline powers are averaged across ipsilateral and contralateral electrodes, (iv) for each window, the difference between ipsilateral and contralateral average powers are normalized by the contralateral power ($\Delta P = \frac{\bar{P}_{ipsi} - \bar{P}_{contra}}{\bar{P}_{contra}}$). This normalization helps to exclude the effect of baseline power variations across time windows and provides a measure of hemispheric power difference, i.e., ΔP , and (v) finally, we perform a non-parametric statistical test using Wilcoxon T-test [9*], to assess the statistical significance of deviation of ΔP from zero. Based on the results, for all of the 12 patients in the dataset, there is a significant difference ($p < 1e-8$) between the ipsilateral and contralateral average power ($\bar{P}_{contra} < \bar{P}_{ipsi}$). The results are consistent with our visual inspection of the baseline powers in ipsilateral and contralateral electrodes and the concept of “breach rhythm” reported in the literature, which is an increase in signal power in a wide range of frequencies in areas with skull defects [6*,7*]. The average power across all patients is 488.2 ± 68.1 (μV^2) and 341.7 ± 39.8 (μV^2) (95% confidence intervals) for ipsilateral and contralateral hemispheres respectively.

Supplementary Fig. X summarizes the results of this analysis. These details are included in the Supplementary Note D.

5. It was pointed out in their previous publication [28] that the WAVEFORM algorithm is not real-time and that future work would extend this algorithm to enable real-time detection – something that was noted as an important factor for clinical applicability. Can the authors comment on processing time required to complete a detection map of SD with the modified WAVEFORM processing described here, and also indicate whether and how significant acceleration towards real-time detection could be expected in the future?

Response to 5) Thanks for the important comments. Real-time monitoring of patients with brain injuries is crucial to predict and prevent worsening brain injuries through SD detections at ICUs. Our modified WAVEFRONT algorithm in this paper can provide SD detection results for each 4-hour time window (epoch) with only ~5min computational delay.

In this paper, we made significant improvements in the speed of SD detection using WAVEFRONT: (i) in the preprocessing pipeline, the ICA analysis is accelerated significantly, by 25 folds, using a GPU implementation of this function (i.e., CUDAICA [10*]). Therefore, preprocessing and pruning a 10-hour scalp EEG recording in this dataset takes minutes (~5 minutes) using a GPU, in comparison to the CPU processing speed (~2 hours), (ii) the main steps of WAVEFRONT are accelerated using parallel processing and distribution across multiple CPU physical cores (64 CPU cores using AMD Ryzen Threadripper PRO 3995WX). E.g., the outlier rejection step is done in parallel for all of the 19 EEG scalp channels and 6 ECoG channels. In addition, SD detection in the main step of the algorithm (including Cylindrical projection, Interpolation and thresholding, Subsampling and optical flow calculation, Quantization of orientations, Orientations bounding boxes (OBBox), and Stitching process and the final decision on detection) is done in parallel for all of the epochs of each patient, up to 64 epochs at the same time (upper bounded by the number of available CPU cores). Once the algorithm is trained and the optimal set of parameters are found, it only takes a few seconds (<5s) to perform SD detection in each epoch of 240min of data. This enables the algorithm to provide SD detection results for each recording epoch, with negligible computational delay (~5min, including the preprocessing time): every 3 hours (epoch step size in our method), WAVEFRONT provides SD detection results. This long detection interval is required to capture the spatiotemporal dynamics of slowly propagating SD waves (1-8 mm/min). The processing performance numbers are provided using a workstation with 64 CPU cores using AMD Ryzen Threadripper PRO 3995WX, 512GB RAM, and using MATLAB R2018b in Windows 10 Enterprise.

To better address this comment, we included these details on the computational performance analysis in Supplementary Note E.

6. Page 14 and Fig 9 show a distribution of estimated speed of propagation of SD events calculated from the WAVEFORM algorithm. It would be useful to compare this with ECoG recordings, but it is noted (p9) that “no ground truth is available for the speed of propagation of annotated SD events in the dataset”. Couldn’t this be calculated from the propagation of events across electrodes of the ECoG strip?

Response to 6-Part1) Thanks for posing this important question. It is not possible to infer the speed of propagation from a single strip of ECoG. The reason is that the linear placement of 6 ECoG electrodes only captures the component of speed of propagation across the strip of electrodes. However, without knowing the angle between the strip and the SD wavefront, the actual speed of propagation cannot be estimated. To better explain this limitation, we show two

examples of SD waves propagating at two different directions (θ_1 and θ_2) and speeds (V_1 and V_2), but with the same speed component V across the strip of ECoG in the following figure:

The reason for this limitation is that the spatial coverage of the strip of ECoG electrodes is limited to a linear array in the brain. Using the strip of 6 electrodes on the cortex, the direction and pattern of propagation of SDs in the brain, and hence the speed, cannot be obtained/estimated reliably. This is a very interesting and important future direction where higher spatial coverage of SD ground truth is available, e.g., a 2D mesh of ECoG electrodes.

This question seems interesting in light of the comment (p13) that “DHC patients have highly concave scalp surface on the hemisphere with missing skull”. It would be helpful to add a comment on whether or not the detected rates and/or patterns of propagation would be as easy to track with WAVEFORM in brain that that better retains the normal structure of sulci and gyri.

Response to 6-Part2) Thanks for this great suggestion. While our comment was about the scalp being concave in these patients, as noted by the reviewer, the structure of the brain is also altered in these patients due to the initial injury, the surgeries (e.g., DHC and hematoma extraction), and brain swelling that often ensues injuries and skull surgeries. We believe that WAVEFRONT, by tracking and detection of individual wavefronts of SDs, enables reliable detection of SDs, even under structural abnormalities in the brain, skull, and scalp. However, to better explore this effect, future studies with higher spatial coverage/resolution of intracranial recording are needed. We acknowledge this as a limitation of the ground truth in this work. We

added the following sentence in the Discussions and Conclusions section following this comment:

“....Notably, the brains of the severe TBI patients in our dataset had structural abnormalities due to the injuries and/or surgical interventions for hematoma extraction, e.g. to alleviate swelling. These structural abnormalities can affect the pattern and speed of propagation of SDs. A ground truth with higher spatial and temporal coverage and accuracy is warranted to further explore these effects and assess the performance of WAVEFRONT...”

7. Per Journal style recommendations, remove use of italics of emphasis.

Response to 7) Thanks. All italics of emphasis are changed to normal font throughout the paper. Only some sub-subtitles are kept in italics, where no emphasis is given.

8. Page 4: Figure 1, the description of red and green arrows should be added to the figure legend, and related more clearly to the text describing the location of DCH and evacuation of hematoma.

Response to 8) Thanks for the great suggestion to improve clarity of Fig. 1. We added the description of the red and green arrows in the caption of this figure and in the text where we talk about this figure.

9. Bottom of page 4/top of page 5: It is stated that a unique SD wave was “annotated by consideration of, and manifested across all electrodes of the ECoG strip”. This should be modified to be consistent with the rest of the paragraph, which describes events classified as SDs that are not seen in all electrodes (or even in a single electrode).

Response to 9) Thanks for catching this error in our explanation of ground truth annotations. To fix this, we modified the sentence as follows:

“In this paper, an SD event refers to a unique SD wave, as annotated by consideration of, and manifested across, all electrodes of the ECoG strip.”

[1*] Rush, S. and Driscoll, D.A., 1968. Current distribution in the brain from surface electrodes. *Anesthesia & Analgesia*, 47(6), pp.717-723.

[2*] Cohen, D. and Cuffin, B.N., 1983. Demonstration of useful differences between magnetoencephalogram and electroencephalogram. *Electroencephalography and clinical neurophysiology*, 56(1), pp.38-51.

[3*] Chamanzar, A., Liu, X., Jiang, L.Y., Vogt, K.A., Moura, J.M. and Grover, P., 2021. Automated, Scalable and Generalizable Deep Learning for Tracking Cortical Spreading

Depression Using EEG. In *2021 10th International IEEE/EMBS Conference on Neural Engineering (NER)* (pp. 416-419). IEEE.

[4*] Chamanzar, A., George, S., Venkatesh, P., Chamanzar, M., Shutter, L., Elmer, J. and Grover, P., 2018. An algorithm for automated, noninvasive detection of cortical spreading depolarizations based on EEG simulations. *IEEE Transactions on Biomedical Engineering*, 66(4), pp.1115-1126.

[5*] Sivakumar, S., Tsetsou, S., Patel, A.B., Stapleton, C.J., Grannan, B.L., Schweitzer, J.S., Chung, D.Y. and Rosenthal, E.S., 2022. Cortical spreading depolarizations and clinically measured scalp EEG activity after aneurysmal subarachnoid hemorrhage and traumatic brain injury. *Neurocritical Care*, 37(Suppl 1), pp.49-59.

[6*] Cobb, W.A., Guiloff, R.J. and Cast, J., 1979. Breach rhythm: the EEG related to skull defects. *Electroencephalography and clinical neurophysiology*, 47(3), pp.251-271.

[7*] Brigo, F., Cicero, R., Fiaschi, A. and Bongiovanni, L.G., 2011. The breach rhythm. *Clinical neurophysiology*, 122(11), pp.2116-2120.

[8*] Chamanzar, A., Behrmann, M. and Grover, P., 2021. Neural silences can be localized rapidly using noninvasive scalp EEG. *Communications Biology*, 4(1), p.429.

[9*] Wilcoxon, F., 1992. Individual comparisons by ranking methods (pp. 196-202). Springer New York.

[10*] Raimondo, F., Kamienkowski, J.E., Sigman, M. and Slezak, D.F., 2012. CUDAICA: GPU optimization of infomax-ICA EEG analysis. *Computational intelligence and neuroscience*, 2012, pp.2-2.

Reviewer #3 (Remarks to the Author):

Review for Communications Medicine

This is an interesting paper that sets out to determine whether there is possible a noninvasive and reliable detection of spreading depolarizations in severe brain injury using a scalp EEG. The authors modified the previously developed WAVEFRONT algorithm and used it to detect spreading depolarizations in 12 patients with severe traumatic brain injury who underwent decompressive hemicraniectomy. The hypothesis was that the automated detection of the SDs using noninvasive electroencephalography (EEG) would be possible. At the same time a SD-detection using ECoG strip electrodes in the decompressive hemicraniectomy side was performed. The quantification of performance was based in the accuracy detection of spreading depolarizations, including true positive rate (TPR), false positive rate (FPR) and the estimation

of the frequency of SDs. The authors conclude that WAVEFRONT can achieve a very good performance in the estimation of the SD detection.

The paper is generally well written and the authors are from a well-known team in the field of neurosurgery and critical care medicine. However, there are some concerns regarding the methodology and presentation of the study. The conclusions are original and convincing. The impact has to be considered as moderate.

The authors would like to thank the reviewer for finding our paper interesting with original and convincing conclusions.

Major points:

1-As mentioned by the authors the small sample size. With only 12 patients is difficult to have a safe conclusion.

Response to 1) We agree that one needs exercise care when performing statistics with limited data. This is precisely why, when we perform cross-validation across patients, we split across patients, and leave two patients out for the validation purpose. The average validation results across all 66 possible validation sets (exhaustive cross-validation technique) in this paper do establish feasibility of detection, but confidence intervals on detection rates are not narrow, e.g., the 95% confidence interval for the average validation accuracy of 74% has 6% length (70.8%-76.7%). Larger datasets would enable stronger statistical guarantees but are also expected to improve the average performance.

The limited sample size, along with other limitations of the dataset (e.g., the limited spatiotemporal resolution and coverage of the ground truth, missing skull, etc.) are fully acknowledged in the manuscript, at various part of the manuscript, including the following sentences in the Discussions and Conclusions:

“....(iii) due to the small number of patients in this study, overfitting to the available SD events is inevitable (see Section III for more details on this issue). We expect WAVEFRONT to achieve a better average validation performance by using a larger dataset of TBI patients with multiple SD events across different varieties of propagation patterns (single-gyrus, semi-planar, ring-shape, etc.), different ranges of propagation speeds, and in different brain regions. In addition, we would be able to provide statistical guarantees for the detection and discrimination results using a larger dataset,...”

2- The fact the authors used only ipsilateral scalp EEG electrodes. May that influence the conclusions?

Response to 2) Indeed, our choice of using only ipsilateral scalp electrodes limits the scope of SD detection to SDs that propagate in the ipsilateral hemisphere, where the skull is at least partially missing. Therefore, the conclusions in this paper on the accuracy of noninvasive SD detection are only validated in the absence of a substantial amount of bone and further studies

on patients with intact skulls are needed. Please see our response to comment 3 of Reviewer#2 for more details on this limitation and future work.

However, it is an important choice: As is discussed in the manuscript, the choice of ipsilateral scalp electrodes for the detection is because of: (i) a limitation of the ground truth labeling, namely, the SD ground truth is only available in the ipsilateral hemisphere using the strip of ECoG placed during the DHC procedure. For the contralateral hemisphere, no ground truth is available in this dataset, which is another limitation of this study, and (ii) the statistical differences between the ipsilateral and contralateral scalp EEG signals (see our response to comment 4 of the second Reviewer for more details). These points are included in Section IIB:

“Excluding the contralateral electrodes is helpful to: (i) tailor the WAVEFRONT algorithm to the SD events that we are certain about (i.e., events that we have a ground truth for) during the training process (see Section III for details), and obtain a more realistic estimation of WAVEFRONT's performance in noninvasive detection of SDs, and (ii) acknowledge the statistical differences between the ipsilateral and contralateral EEG signals, as ignoring these differences may adversely affect the performance of WAVEFRONT (see Supplementary Note D for hemispheric comparison of average baseline power on the scalp). Because EEG signals tend to be less sensitive to contralateral sources, we expect this restriction to not hurt the performance of our algorithm.”

To address this comment, we added the following sentence in the Discussions and Conclusions:

“...In this paper, we only validated the performance of our technique using ipsilateral electrodes in the hemisphere with removed skull, which limits the conclusion of this study to patients with DHC...”

3-The performance of the WAVEFRONT to detect cluster of SDs is not clear.

Response to 3) Thank you for this comment. Clustered SDs are defined as more than two SDs in a time interval of 3 hours or less [1*,2*]. Our results strongly suggest that WAVEFRONT underdetects SDs when they are highly clustered, and this is a significant limitation that future work could address. This issue is discussed in detail in Section III, Results, under the subsection of “Performance of WAVEFRONT in prediction of SD frequency”, and illustrated in Fig. 6, 8, and 9. The most informative figure that shows this underdetection is Fig. 6, where the algorithm detects 5 individual SD events (shown in the ECoG traces in red) as 3 separate detection intervals (blue detection bars on the bottom). As discussed in the paper, this can be in part due to the inherent limitation of EEG and/or limitation of our method in underdetection of highly clustered SDs.

- [1*] S. Jewell et al. Development and evaluation of a method for automated detection of spreading depolarizations in the injured human brain. *Neurocritical care*, 35(2):160–175, 2021.
- [2*] D. Hertle et al. Effect of analgesics and sedatives on the occurrence of spreading depolarizations accompanying acute brain injury. *Brain*, 135(8):2390–2398, 2012.

REVIEWERS' COMMENTS:

Reviewer #1 (Remarks to the Author):

The authors addressed all the comments appropriately. It can be accepted.

Reviewer #2 (Remarks to the Author):

This revision addresses the comments raised by this reviewer in a good degree of detail. The main body of the manuscript has been shortened, and large amount of technical information has been moved to a supplementary section.

While the introduction is shorter and benefits from having less methodological detail, it still does not finish with a brief summary of the major results and conclusions. The last paragraph of the discussion instead is a summary of the structure of the manuscript, rather than clearly stating the major new results and conclusions.

The information that has been placed into the series of Supplemental notes is includes helpful technical details, as well as other points of discussion and clarification of points raised in response to the reviewers' comments. Some of this is helpful to streamline the flow of the methods section in the main document, but the scattering of other information (including responding to reviews) does not make it easy for the reader. For example, Supplementary Note A contains detailed discussion of the conclusions and limitations of the study, and this needs to be integrated by the reader with the other Discussion & Conclusion section that already exists within main document. The main points should instead be consolidated and presented in one place (the main document). A second example relates to a point raised in the initial review, regarding effects of craniectomy on EEG power. This clarification is needed within the description of "Epoching and envelope extraction" methods (now Supplementary Note C). However the clarification isn't mentioned in that section, and the response to the critique is instead added as a new stand-alone Supplementary Note D. The main conclusion from the new population analysis should be mentioned in the appropriate methods description. Another example related to the initial review is the question of processing time for clinical utility. The first paragraph of Supplementary Note E addresses this question, and inclusion of this main point in the main body of the manuscript would likely increase impact for many readers.

Reviewer #3 (Remarks to the Author):

The revised version of the manuscript should be published in Communication Medicine.

Noninvasive, automated and reliable detection of spreading depolarizations in severe traumatic brain injury using scalp EEG

Alireza Chamanzar, Jonathan Elmer, Lori Shutter, Jed Hartings, Pulkit Grover

Response letter cover:

We thank the reviewers and the editor of our paper for the positive feedback and constructive comments they provided through the peer review process. We are excited about the possibility of acceptance of our revised paper. In this revised version of the manuscript, we attempted to fully address the remaining concerns raised by the reviewers. We are grateful for the reviewers' feedback, which made the content of the paper easier to understand and follow. Following are our point-by-point responses to the reviewers' comments:

Color codes:

- Reviewers' comments/questions
- Authors' responses
- **Bold fonts: Modified/added texts in the revised manuscript**

Reviewers' comments:

Reviewer #1 (Remarks to the Author):

The authors addressed all the comments appropriately. It can be accepted.

We thank the reviewer for the positive assessment of the manuscript.

Reviewer #2 (Remarks to the Author):

This revision addresses the comments raised by this reviewer in a good degree of detail. The main body of the manuscript has been shortened, and large amount of technical information has been moved to a supplementary section.

1. While the introduction is shorter and benefits from having less methodological detail, it still does not finish with a brief summary of the major results and conclusions. The last paragraph of the discussion instead is a summary of the structure of the manuscript, rather than clearly stating the major new results and conclusions.

Response to 1) We thank the reviewer for this important comment about the structure of the introduction. To address this comment and end the introduction with a summary of the major results and conclusions, we removed the last paragraph and moved the discussion on the clinical relevance of our work to an earlier part of the introduction. In the revised manuscript, the

last two paragraphs of the introduction include the highlights of the results and conclusions on the feasibility of noninvasive SD detection:

“Clinical relevance of noninvasive SD detection in patients with DHC. As discussed earlier, this work focuses on severe TBI patients with DHC. A natural question is whether noninvasive and automated SD detection in patients with removed skull parts is clinically relevant? The DHC procedure is part of the standard of care for many severe TBI patients³⁴ to control elevated intracranial pressure, extract hematoma, and prevent further damage to the brain tissue^{35–37}. It is worth noting that following a DHC, the scalp is sutured back over the brain, even though a piece of the skull is missing. Patients who receive DHC are continuously monitored in the ICU after their scalp incision is closed. During this period, while intracranial monitoring of SDs can provide higher spatial resolution⁸, scalp EEG-based automated SD detection can provide valuable clinical information pertaining to worsening brain injury. ~~Intracranial monitoring of SDs can provide more information than the presence or absence of these waves in the brain, such as characteristic patterns of SDs, including clustered or persistent depressions of spontaneous neural activities⁸. However,~~ Scalp EEG has broader spatial coverage than a locally placed ECoG strip. It also provides better spatial resolution in DHC patients compared to the intact skull EEG recordings³⁵, at least close to locations where the skull has been removed. Further, while procedural risks (e.g., bleeding, infection, etc.) associated with subdural electrodes are infrequent^{38,39}, noninvasive EEG precludes their possibility entirely. Therefore, noninvasive SD detection in severe TBI patients with DHC can prove clinically valuable in improving outcomes.

In Results, we show that WAVEFRONT achieved a reliable SD detection performance using Delta band EEG recording, with a ~74% average true positive rate (TPR) and less than 1.5% false positive rate (FPR) using cross-validation, ~~as illustrated in Fig. 4. Such a high TPR attained with a low FPR resolves the feasibility question: noninvasive SD detection is possible, at least for patients who have received DHC. However, is this performance sufficient for clinical goals? To answer this question, we performed an additional analysis, predicting the number of SDs from the total minutes of detected SD events. This analysis was inspired by Jewell et al.’s¹⁰ estimation of SDs’ frequency; unlike in our study, their aim was to automate invasive detection of SDs (details of this work are in Section). Our preliminary results, albeit with limited data, suggest that WAVEFRONT can reliably estimate the number of SD occurrences in long 30-hour time intervals using a regression analysis, ~~with $R^2 \sim 0.71$, as is shown in Fig. 7 and explained in detail in.~~ Overall, we believe that WAVEFRONT’s performance indicates that noninvasive prognostication of worsening brain injury using SD detection is possible. However, to understand this potential, further studies with more data are warranted.~~

~~In Section II, we provide a detail description of the dataset and SD ground truth labeling. In addition, we describe the EEG preprocessing pipeline and the WAVEFRONT algorithm, along with the modifications and improvements we made. In Section III, we present the performance of WAVEFRONT in noninvasive SD detection across different EEG~~

~~frequency bands and evaluate the frequency of SDs in large time windows. Finally, we conclude in Section IV, where we discuss some of the limitations of our study and discuss future work's promising direction. A more extensive discussion on the limitations and future work are included in Supplementary Note A.~~

2. The information that has been placed into the series of Supplemental notes includes helpful technical details, as well as other points of discussion and clarification of points raised in response to the reviewers' comments. Some of this is helpful to streamline the flow of the methods section in the main document, but the scattering of other information (including responding to reviews) does not make it easy for the reader. For example, Supplementary Note A contains detailed discussion of the conclusions and limitations of the study, and this needs to be integrated by the reader with the other Discussion & Conclusion section that already exists within main document. The main points should instead be consolidated and presented in one place (the main document).

Response to 2) To address this comment, we removed Supplementary Note A and combined its content with the discussions in the main manuscript, in Discussions. In addition, we pushed some of the details on the modifications and improvements of WAVEFRONT to Method, where we present the details on the algorithm steps.

The following paragraph is now added in Discussion:

“Validation of WAVEFRONT on DHC patients is a good starting point for noninvasive and automated SD detection because: a) intracranial ground truth for SDs can be obtained by placing ECoG electrodes during the DHC procedure, and b) head layers, including skull, meninges, cerebrospinal fluid (CSF), and scalp, have low-pass filtering or “blurring” effects on the scalp EEG signals. This makes the detection and tracking of narrow SD waves challenging^{29,30}. This is less challenging in DHC patients due to the absence of the low-conductivity skull layer. Nevertheless, the challenge is considerable: i) relative to ECoG, the signal is more noisy and spatially low-pass filtered, and ii) as the SD wave propagates into the sulcus, its representation in the scalp EEG signals reduces substantially. This breaks the waves, as measurable by EEG, into disconnected components, which we call “wavefronts”³⁰. Complex patterns of SDs (e.g., single gyrus^{61,62}, semi-planar^{62–64}, etc.) can make the noninvasive detection of these waves even more challenging. WAVEFRONT addresses some of the difficulties in noninvasive detection of SD waves in EEG. It breaks down the challenging task of detecting the whole propagating SD wave in the brain using noisy and blurry filtered scalp EEG signals into simpler tasks of detection and classification of disjoint SD wavefronts. This overcomes the challenge related to the effects of sulci and gyri discussed above and enables the detection and tracking of complex patterns of propagation.”

And the following paragraphs are added at the beginning of “Methods: SD detection and tracking using WAVEFRONT”:

“We used our previously proposed WAVEFRONT SD detection algorithm³⁰, with appropriate modifications and improvements, to detect and track SD waves in EEG recordings of 12 TBI patients in the dataset (see Table I for more details on these patients). WAVEFRONT is an explainable automated SD detection framework with intuitive steps and interpretable detection outputs and results. It addresses the challenges of noninvasive detection of SD waves in EEG (see Discussion for details) by breaking down the challenging task of detecting the whole propagating SD wave in the brain using noisy and blurry filtered scalp EEG signals into simpler tasks of detection and classification of disjoint SD wavefronts, following these steps: Power envelopes of the scalp EEG signals at each electrode are extracted, and depressions (power reductions) are detected. These detected depressions are then projected onto a 2D plane to obtain depression wavefronts. Propagating SD wavefronts are then detected and tracked based on their speed and direction of propagation. To estimate the speed and direction of propagation of depression wavefronts on these 2D planes, WAVEFRONT uses a computer vision technique called optical flow^{46,47}. It then stitches together the detection of these wavefronts over time and space to detect and track SD waves in the brain. This overcomes the challenge related to the effects of sulci and gyri and enables the detection and tracking of complex patterns of propagation.

Although the simulation results of automated SD detection in³⁰ are promising, we recognize that WAVEFRONT suffers from certain technical shortcomings and cannot be directly applied to real scalp EEG signals for SD detection: (i) it uses a fixed set of parameters (e.g., depression level/depth threshold, temporal score threshold, and spatiotemporal neighborhood radius); (ii) it is highly sensitive to the amplitude outliers; (iii) it implicitly assumes the power level of normal background brain activity (DC offset of the power envelope) is stable and not changing over time (see Fig. 9 and 10 in³⁰ for more details), which limits the ability of WAVEFRONT in the detection of depressions, as well as near-DC shifts during propagating SDs in the real EEG recordings; (iv) it does not address the challenges of using a low-density EEG grid, including the high rate of false alarms due to the non-propagating depressions on the scalp that we observe here; (v) it does not address the challenges of using a very low-density EEG grid, including the high rate of false alarms due to the non-propagating depressions on the scalp; and (vi) it estimates the optical flows in pixels on the 2D images, rather than in terms of the physical distances on the scalp, which can introduce errors in estimation of the speed and direction of propagation of SD wavefronts. In this work, we addressed these limitations of WAVEFRONT by making necessary modifications and improvements, including designing a training and validation framework for the algorithm to learn an optimal set of parameters through a crossvalidation analysis. Other modifications include (a) designing a rigorous and automated preprocessing pipeline for outlier rejection and pruning the EEG signals, (b) using a power-envelope extraction method that is less sensitive to large-amplitude artifacts (i.e., outliers), (c) extending the depression extraction method to be able to detect DC shifts in the near-DC components (1–10 mHz) as well as the power depressions in the higher frequency bands (≥ 0.5 Hz), (d) defining an “effective propagation measure” along with a learnable threshold on this

measure to reject the false alarms of the non-propagating depressions on the scalp, and (e) mapping the estimated optical flows on the scalp spherical surface.... ”

3. A second example relates to a point raised in the initial review, regarding effects of craniectomy on EEG power. This clarification is needed within the description of “Epoching and envelope extraction” methods (now Supplementary Note C). However the clarification isn’t mentioned in that section, and the response to the critique is instead added as a new stand-alone Supplementary Note D. The main conclusion from the new population analysis should be mentioned in the appropriate methods description.

Response to 3) Following this suggestion, we added a clarification statement on the hemispheric baseline power analysis in Methods, subsection “Epoching and envelope extraction”:

“As discussed in Methods, we only used ipsilateral scalp EEG electrodes for each patient because of the missing spatial SD ground truth in the contralateral hemisphere, and heterogeneity of the baseline EEG power between the hemispheres with DHC and the hemisphere with an intact skull. **Based on the results, for all of the 12 patients in the dataset, there is a statistical difference ($p < 1e-8$) between the ipsilateral and contralateral average power (see Supplementary Note 1 for hemispheric comparison of average baseline power on the scalp) ...”**

4. Another example related to the initial review is the question of processing time for clinical utility. The first paragraph of Supplementary Note E addresses this question, and inclusion of this main point in the main body of the manuscript would likely increase impact for many readers.

Response to 4) Thanks for this suggestion to increase the impact of our manuscript. We have now included the main point on the processing time for clinical applications in the main manuscript, in Discussion:

“Real-time monitoring of patients with brain injuries is crucial to predict and prevent worsening brain injuries through SD detection at ICUs. Our modified WAVEFRONT algorithm in this paper can provide SD detection results for each 4-hour time window (epoch) with only 5min computational delay (see Supplementary Note 2 for details).”

Reviewer #3 (Remarks to the Author):

The revised version of the manuscript should be published in Communication Medicine. We thank the reviewer for the positive assessment of the manuscript.